# Mucosal sugars delineate pyrazine vs pyrazinone autoinducer signaling in *Klebsiella oxytoca*

Randy Hamchand [1,2], Kevin Wang [2], Deguang Song[3], Noah W. Palm [3] & Jason M. Crawford [1,2,4] ✉

Virulent *Klebsiella oxytoca* strains are associated with gut and lung pathologies, yet our understanding of the molecular signals governing pathogenesis remains limited. Here, we characterized a family of *K. oxytoca* pyrazine and pyrazinone autoinducers and explored their roles in microbial and host signaling. We identified the human mucin capping sugar Neu5Ac as a selective elicitor of leupeptin, a protease inhibitor prevalent in clinical lung isolates of *K. oxytoca*, and leupeptin-derived pyrazinone biosynthesis. Additionally, we uncovered a separate pyrazine pathway, regulated by general carbohydrate metabolism, derived from a broadly conserved PLP-dependent enzyme. While both pyrazine and pyrazinone signaling induce iron acquisition responses, including enterobactin biosynthesis, pyrazinone signaling enhances yersiniabactin virulence factor production and selectively activates the proinflammatory human histamine receptor H4 (HRH4). Our findings suggest that the availability of specific carbohydrates delineates distinct autoinducer pathways in *K. oxytoca* that may have differential effects on bacterial virulence and host immune responses.

K*lebsiella oxytoca* is a gram-negative, facultative anaerobe that colonizes mucus-rich organs, such as the lungs and gut[1–5]. The impact of *K. oxytoca* on human physiology is strain dependent, with some acting as human commensals associated with colonization resistance against pathogenic bacteria[6], and others as opportunistic pathogens associated with nosocomial respiratory infections[1,2,7] and gut disease[3,4]. The role of pathogenic *K. oxytoca* strains in gut disease, namely antibiotic-associated hemorrhagic colitis, is attributed to its resistance to ß-lactam antibiotics, which enables it to outcompete competitors during ß-lactam challenge, and its ability to produce the pyrrolobenzodiazepine enterotoxins tilivalline and tilimycin[8,9]. Interestingly, the genotypes of *K. oxytoca* isolates from patients with nosocomial respiratory infections are distinct from those from patients with hemorrhagic colitis[10]. Indeed, the biosynthetic gene cluster (BGC) responsible for the production of tilivalline and tilimycin correlates with colitis but not

with nosocomial respiratory infections in a panel of human clinical respiratory and fecal isolates[10,11]. Rather, a BGC responsible for leupeptin biosynthesis correlates with respiratory isolates of patients with varying lung pathologies, including pharyngitis, exacerbated chronic obstructive pulmonary disease, and ventilator-associated pneumonia[12].

In contrast to other human pathogens, relatively little is known regarding the factors that regulate the *K. oxytoca* metabolome and transcriptome. Generally, for bacteria to respond effectively to external cues and nutritional requirements, they develop intricate cell-to-cell communication systems that enable them to sense and adapt to their environment[13,14]. These responses are often mediated through extracellular small molecule signaling compounds known as autoinducers, through a process called quorum sensing. Quorum sensing is a population-dependent mechanism that serves a critical role in

[1]Department of Chemistry, Yale University, New Haven, CT, USA. [2]Institute of Biomolecular Design & Discovery, Yale University, West Haven, CT, USA. [3]Department of Immunobiology, Yale University School of Medicine, New Haven, CT, USA. [4]Department of Microbial Pathogenesis, Yale University School of Medicine, New Haven, CT, USA. ✉e-mail: jason.crawford@yale.edu

coordinating gene expression changes in bacteria that lead to synchronized behaviors such as biofilm formation and the production of virulence factors and secondary metabolites[15–23].

Most extant literature on *Klebsiella* quorum sensing focuses on the human respiratory pathogen *K. pneumoniae*[24–26]. *K. pneumoniae* encodes a homologue of the *luxS* gene, which is involved in the biosynthesis of autoinducer 2 (AI-2)[27]. AI-2, which was first discovered from *Vibrio harveyi*[28], is found in many bacteria and is considered an interspecies signaling molecule[29–32] that plays a key part in regulating biofilm production[32–34]. Like *K. pneumoniae*, *K. oxytoca* genomes also contain a homologue of the *luxS* gene. In addition to AI-2, *Klebsiella* isolates produce dimethylpyrazine-2-ol (DPO) and autoinducer-3 (AI-3) analogs[35,36]. DPO, AI-3, and AI-3 analogs are pyrazinone quorum sensing signals produced by a variety of bacteria that are derived from threonine dehydrogenase (Tdh)-dependent aminoacetone production. DPO administration has been shown to regulate biofilm production in *Vibrio cholerae* while AI-3 and some of its analogs regulate virulence expression in enterohemorrhagic *Escherichia coli*[35,36].

In this study, we investigated the conditions necessary to elicit transcription of the leupeptin BGC in *K. oxytoca*, which led to the identification of specific carbohydrate nutrients that activate the leupeptin pathway and a separate previously undescribed family of pyrazine autoinducers. We found that the leupeptin BGC is selectively activated upon media supplementation with the human mucin-capping sugar *N*-acetylneuraminic acid (Neu5Ac) while pyrazine biosynthesis is dependent on carbohydrate metabolism more broadly. Activation of the leupeptin BGC leads to the production of an Arg-Leu derived pyrazinone that shares the same core scaffold of DPO and AI-3. Meanwhile, the pyrazine pathway proceeds through an aminoketone species that resembles DPO and AI-3 analogue precursors. We probed the biosynthesis of the *Klebsiella* pyrazines and found that their production requires a pyridoxal phosphate (PLP)-dependent enzyme, deemed Pyr (for pyrazine biosynthesis protein), which facilitates the coupling of tyrosine or phenylalanine with succinyl-coenzyme A to form δ-amino acids. These δ-amino acids homodimerize, heterodimerize with each other, or heterodimerize with aminoacetone (AA), a product derived from Tdh, to form the *Klebsiella* pyrazines. Phylogenetic analysis and sequence similarity networking revealed that the *pyr* biosynthetic gene is broadly distributed across bacterial taxa, and heterologous expression of Pyr homologs in *E. coli* confirmed that they produce *Klebsiella* pyrazines. Through RNA-sequencing experiments, we show that the *Klebsiella* pyrazines rewire metabolism and promote transcription of general iron acquisition pathways, while the leupeptin pyrazinone similarly upregulates iron acquisition and metabolism pathways, including the virulence factor yersiniabactin. Furthermore, through PRESTO-Tango GPCR analysis[37], we found that the leupeptin pyrazinone is a selective agonist of histamine receptor H4, a human GPCR involved in asthma and allergic disorders, while the *Klebsiella* pyrazines are not. Together, these findings suggest distinct autoinducer pathways in *K. oxytoca*, that are mediated by the availability of specific carbohydrates, regulate differential effects on bacterial metabolism, iron acquisition, and host responses.

## Results

### Neu5Ac elicits leupeptin biosynthesis in *Klebsiella oxytoca*

We previously examined a panel of 84 clinical *K. oxytoca* isolates (including recently renamed *Klebsiella michiganensis* isolates) for the presence of either the *leup* operon or *npsB*, an NRPS of the tilimycin/tilivalline gene cluster[12]. We found that the presence of the leupeptin BGC in *K. oxytoca* correlates with nosocomial respiratory infections in this panel (14 out of 19 lung isolates, 2 out of 23 stool isolates), while the *npsB* gene correlates with intestinal tract dysfunction as expected (21 out of 23 stool isolates, 3 out of 19 lung isolates)[12]. While the roles of tilivalline and tilimycin in hemorrhagic colitis are well studied[8,9], the operon for leupeptin biosynthesis is "silent" in *K. oxytoca* under typical

laboratory culture conditions (i.e., leupeptin production is undetectable). Therefore, our study commenced by exploring growth conditions required to trigger leupeptin biosynthesis in *K. oxytoca*. To achieve this, we generated a green fluorescent protein (GFP)-based reporter system to monitor *leup* transcription in *K. oxytoca* ATCC 8724 (recently renamed as *Klebsiella michiganensis* (ATCC 8724), Fig. 1a). We utilized λ-red recombineering techniques[38] to replace the first biosynthetic gene (*leupA*) of the leupeptin BGC with a GFP-spectinomycin resistance cassette. Notably, while the *spec* gene has its own promoter, the *gfp* gene in this design is regulated by the promoter for the leupeptin BGC. Consequently, conditions that elicit transcription of the leupeptin BGC can be monitored by GFP fluorescence intensity.

To determine conditions that promote leupeptin biosynthesis, we adopted a combination of OSMAC (one strain many compounds) and host sensing approaches for metabolite activation[39,40]. This involved culturing wildtype *K. oxytoca* and the reporter strain in a variety of media, including specialized host components (Fig. 1b, top). Porcine mucin broth emerged as a potential weak elicitor of leupeptin biosynthesis in the fluorescence-based screen. Indeed, the presence of porcine mucin in the cultivation medium led to a low but detectable signal by high-performance liquid chromatography quadrupole time-of-flight mass spectrometry (HPLC-QTOF-MS, Supplemental Information), whereas all other media investigated exhibited no detectable signal. This discovery aligns with previous findings suggesting that mucin can act as a nutrient source and signal for differential gene expression in bacteria like *Acinetobacter baumannii*[41]. In bacteria such as *Clostridioides difficile* and *Salmonella enterica* serovar Typhimurium, this alteration in gene expression is mediated by bacterial utilization of discrete carbohydrates released during mucin metabolism[42]. Therefore, we investigated an array of sugars for their ability to activate the reporter strain (Fig. 1b, bottom). We identified *N*-acetylneuraminic acid (Neu5Ac), the capping sugar of human mucins, as a key signal for *leup* transcription. Addition of Neu5Ac to the cultivation medium led to transcriptional activation in the GFP reporter and robust detection of leupeptin by HPLC-QTOF-MS. Neu5Ac and *N*-glycolneuraminic acid (Neu5Gc) are the predominant sialic acids found in mammals[43]; however, humans cannot produce Neu5Gc due to the inactivation of endogenous cytidine monophospho-*N*-acetylneuraminic acid hydroxylase[44]. Unlike Neu5Ac, Neu5Gc was unable to induce the production of **1** (Supplemental Information). Interestingly, *K. oxytoca* lacks the sialidase known to digest the capping sugar of mucin; however, studies have shown that pathogenic bacteria without sialidase enzymes can exploit commensal mucin-degrading bacteria to promote their growth through Neu5Ac metabolism[42].

Next, we performed automated tandem mass spectrometry (MS²) on wildtype *K. oxytoca* with and without Neu5Ac and searched for differentially regulated molecular families. Consistent with our reporter and HRMS analyses, we identified a molecular family that contained features that matched leupeptin (**1**, $[M + H]_{theoretical} = 427.3027$ Da; $[M + H]_{obs} = 427.3002$ Da) and a previously reported lysine analog of leupeptin (**2**, $[M + H]_{theoretical} = 399.2966$ Da; $[M + H]_{obs} = 399.2989$ Da) (Fig. 1c, d; Supplemental Information)[12]. This molecular family contained 19 molecular features, which were interpreted as derivatives of either **1** or **2** (Fig. S1a).

To ascertain whether Neu5Ac operates as a nutritional or chemical signal towards the production of the leupeptins, we generated a *K. oxytoca nanA* mutant. The Nan operon (Fig. S1b) is responsible for metabolizing sialic acids like Neu5Ac in various bacteria including *Escherichia coli*[45], *Corynebacterium diphtheriae*[46], and *Pasteurella multocida*[47]. Comparing the production of **1** between wildtype and Δ*nanA K. oxytoca* strains when supplemented with Neu5Ac revealed that **1** was exclusively produced in the wildtype, suggesting that metabolism of Neu5Ac is required for activation of the *leup* BGC (Fig. 1e). Thus, we refer to Neu5Ac as a "nutrient signal" in this context.

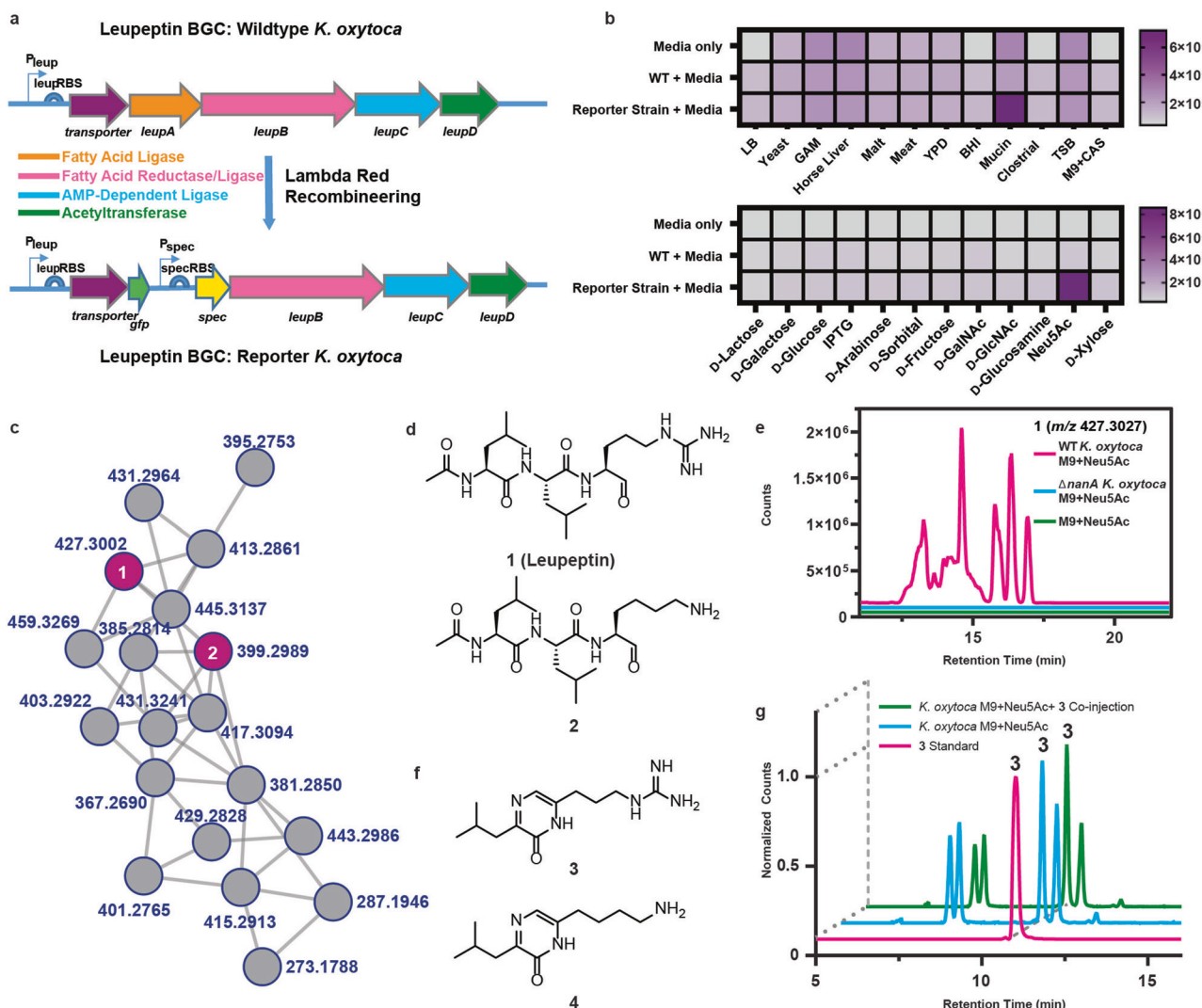

**Fig. 1 | Elicitation of the leupeptin biosynthetic gene cluster in *Klebsiella oxytoca*. a** Design of the GFP-based transcriptional reporter system for the *leup* operon. **b** GFP fluorescence monitoring of *K. oxytoca* strains when cultured in different media (top) and carbohydrate sources (bottom). Fluorescence was measured in relative fluorescence units (RFUs) **c** Molecular networking analysis of emergent *K. oxytoca* leupeptin molecular family when supplemented with Neu5Ac. Connected nodes show a high degree of similarity in their MS² fragmentation patterns. **d** Chemical structures of the major *leup* biosynthetic products leupeptin (**1**) and a lysine-leupeptin analog (**2**) identified from Neu5Ac supplemented *K. oxytoca* cultures. **e** Representative HPLC-QTOF-MS traces (*n* = 3) of the metabolism-dependence of Neu5Ac on the ability of *K. oxytoca* to produce leupeptin. Broad chromatography of leupeptin is due to racemization, hydration, and cyclization events in equilibrium at the arginal residue. **f** Chemical structures of *leup*-derived pyrazinones **3** and **4**. **g** HPLC-QTOF-MS analysis of Neu5Ac-supplemented *K. oxytoca* extracts, synthetic **3**, and their co-injection. Shown are extracted ion chromatograms for the exact mass of **3** (*m/z* 252.1819) +/− 10 ppm.

Notably, the *nanA* mutant was unable to produce **1** when cultivated in mucin medium, supporting the claim that Neu5Ac is a specific elicitor of the *leup* operon (Supplemental Information).

In an earlier investigation on the biosynthesis of leupeptin, we uncovered that the *leup* BGC from *Photorhabdus luminescens* was responsible for the production of a leupeptin (**1**)-derived leucine-arginine pyrazinone (**3**, Fig. 1f)[12]. Mining our metabolomics data, we identified a feature consistent with pyrazinone **3**, along with a leucine-lysine pyrazinone (**4**, Fig. 1f), which could result from the proteolytic cleavage, cyclization, and oxidation of **2** (Supplemental Information). We confirmed the presence of **3** in Neu5Ac-supplemented *K. oxytoca* cultures through chromatographic comparison of culture extracts to a synthetic standard (Fig. 1g).

## Characterization of a family of pyrazines from *K. oxytoca*

In addition to the production of the *leup* products **1**–**4**, metabolomic analysis of Neu5Ac-supplemented wildtype and Δ*nanA K. oxytoca* cultures revealed additional molecular features that were

enriched by Neu5Ac metabolism (Fig. 2a). Molecular networking of the MS² spectra of these features indicated that most of the compounds belonged to an uncharacterized molecular family separate from the leupeptins, with some structural outliers (Fig. 2b)[48]. To gain structural insights into these unknown metabolites, we compared the MS² spectra of all the features against the Global Natural Products Social (GNPS) Molecular Networking online repository[48]. Although this search did not lead to the identification of any features from the molecular network in Fig. 2b, it suggested that the major outlier compounds were related to yersiniabactin: a siderophore virulence factor known to enhance bacterial pathogenicity via its iron and metal acquisition properties[49]. Specifically, the compounds differentially regulated between the wildtype and Δ*nanA K. oxytoca* strains were identified as the yersiniabactin diastereomers (**10a, 10b**), the yersiniabactin-iron complex (**10-Fe**), and a yersiniabactin shunt product (**11**)[50]. To confirm whether these compounds were derived from the yersiniabactin pathway, we generated a *K. oxytoca* mutant (Δ*yer::spec*) which replaced the high

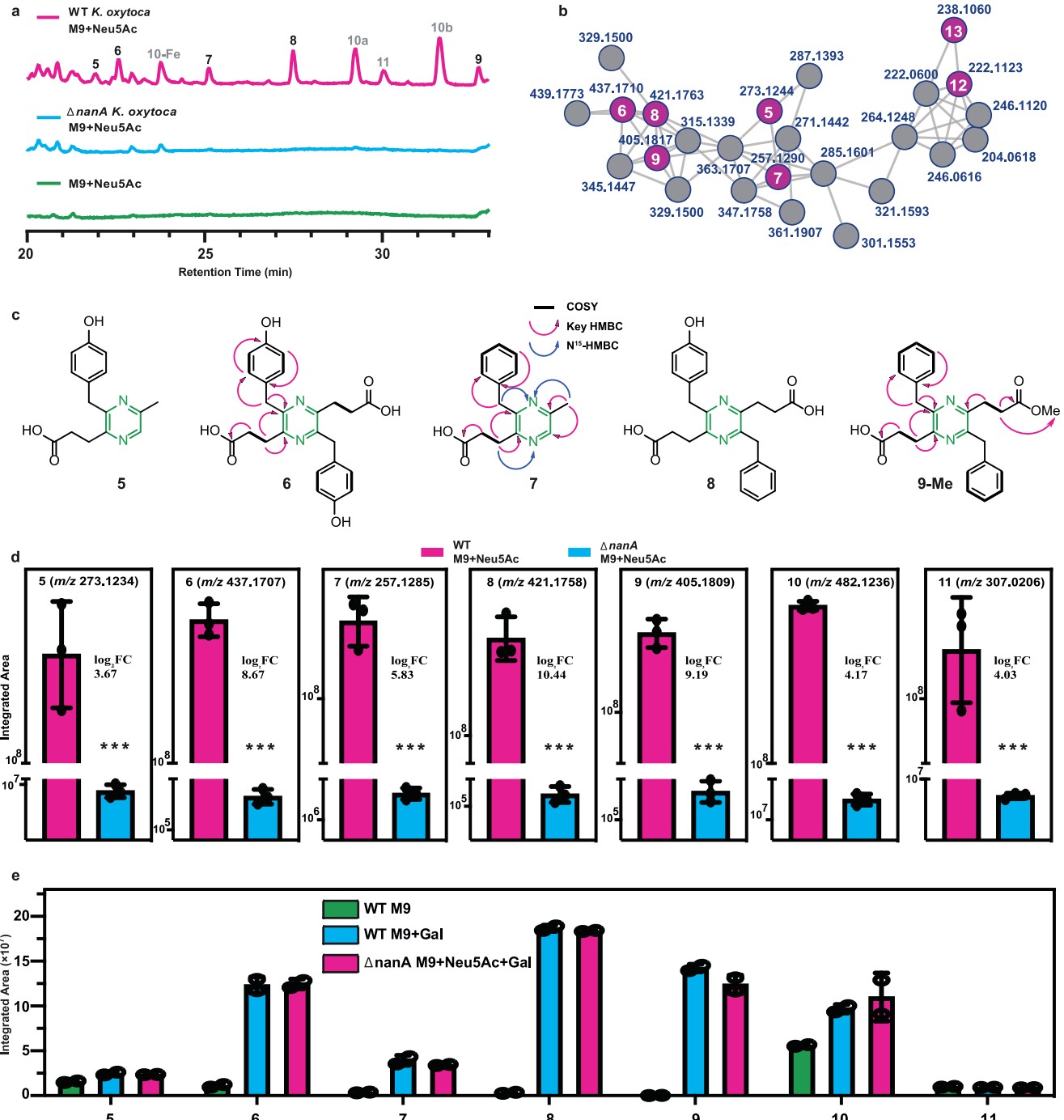

**Fig. 2 | Carbohydrate metabolism promotes pyrazine biosynthesis in *K. oxytoca*. a** Representative HPLC-QTOF-MS total ion chromatograms (*n* = 3 biological replicates) of wildtype *K. oxytoca*, Δ*nanA K. oxytoca*, and medium control samples supplemented with Neu5Ac. Pyrazine compounds are numbered in black; yersiniabactin compounds are numbered in gray. **b** Molecular networking analysis of the pyrazine molecular family. Connected nodes share high degrees of similarity in their MS² fragmentation patterns. **c** Molecular structures of *K. oxytoca* pyrazines. Key NMR signals are indicated for isolated pyrazines. Pyrazine scaffolds are highlighted in green. **d** Relative abundances of identified pyrazines between wildtype *K. oxytoca* and Δ*nanA K. oxytoca* supplemented with Neu5Ac (*n* = 3 biological replicates). Pyrazines were not detected in the growth medium control samples. P-values for the production of all compounds between wildtype and Δ*nanA* groups are < 0.0001. **e** Relative production of pyrazines **5**−**9** and yersiniabactins **10** and **11** in wildtype and Δ*nanA K. oxytoca* strains (*n* = 2 biological replicates) when supplemented with D-Galactose or Neu5Ac and D-Galactose, respectively. Representative **10b** was chosen for integration. Statistical analyses performed through an unpaired two-tailed t-test; n.s. indicates a non-significant difference, * indicates *p* < 0.05; ** indicates *p* < 0.01, and *** indicates *p* < 0.001. Data are represented as mean values +/− SD.

molecular weight protein (HMWP2) gene of the yersiniabactin pathway with a spectinomycin-resistance cassette (Fig. S2a). Indeed, the Δ*yer* strain abrogated production of the yersiniabactin products **10a**, **10b**, **10-Fe**, and **11** (Supplemental Information).

To determine the remaining structures in the molecular network (Fig. 2b), we cultivated 5 liters of wildtype *K. oxytoca* and utilized preparative-scale high-performance liquid chromatography (HPLC) to purify and characterize the major features from the unknown

molecular family. All of the features within the molecular family shared a distinct chromophore with a broad absorption profile from 250 nm to 320 nm and a $\lambda_{max}$ at 284 nm (Supplemental Information). Therefore, we employed a combination of chromophore- and mass-directed isolation techniques to purify the more abundant compounds. Structural analyses of the isolated compounds (**6**, **7**, and **9-Me**) by 1D- ($^1$H) and 2D-nuclear magnetic resonance (NMR) spectroscopy (gCOSY, gHSQCAD, gHMBCAD) were consistent with the family containing a pyrazine scaffold (Fig. 2c). Carboxylic acid methyl esterification of several features (e.g., **9-Me**) was observed during purification (i.e., Fischer esterification of the non-methylated metabolite), which is a finding consistent with the methanolic conditions used during sample processing. Our observation that the purified pyrazines contained succinyl, benzyl, p-hydroxybenzyl, and methyl modifications enabled us to propose structures for two additional metabolites in the molecular family: **5** and **8** (Fig. 2c).

In contrast to compounds **1–4**, which were undetectable in the *nanA* mutant, pyrazines **5–9** and yersiniabactins **10** and **11** were detectable in this strain when supplemented with Neu5Ac (Fig. 2d). Notably, the production of compounds **5–11** dramatically decreased, but were not abolished, in the Δ*nanA* strain relative to wildtype when supplemented with Neu5Ac. This finding suggests that the production of **5–11** may be regulated by more diverse carbohydrate metabolism than the leupeptins. Initiation of yersiniabactin biosynthesis requires salicylic acid, which is produced through the shikimic acid pathway (Fig. S2a)[51,52]. The shikimic acid pathway, in turn, relies on sugar catabolism, as it requires phosphoenolpyruvate and D-erythrose-4-phosphate, both of which are products of carbohydrate metabolism[52]. As such, carbohydrate metabolism could directly affect yersiniabactin production through substrate availability.

To test the hypothesis that Neu5Ac metabolism was not required for production of metabolites **5–11**, we cultured wildtype *K. oxytoca* with or without D-galactose supplementation and Δ*nanA K. oxytoca* with Neu5Ac and D-galactose, and assessed the production of compounds **5–11** (Fig. 2e). In the wildtype strain, the addition of exogenous D-galactose increased the production of compounds **5–10**, with no impact on the production of the yersiniabactin shunt product **11**. Furthermore, complementing Neu5Ac supplemented Δ*nanA* strains with D-galactose also resulted in increased production of compounds **5–10** comparable to D-galactose supplementation alone. This result supports our hypothesis that pyrazine and yersiniabactin production are affected by carbohydrate metabolism more broadly than leupeptin pyrazinone production. We further cultured wildtype *K. oxytoca* in the presence of various sugar sources and observed appreciable quantities of **5–11** for almost all screened conditions (Fig. S2b). Neu5Ac supported the highest observed level of yersiniabactin among the sugar panel, with the exception of lactose which could not induce pyrazine formation. These data suggest that diverse sugars support pyrazine and siderophore biosynthesis, whereas Neu5Ac is specifically required for leupeptin biosynthesis in *K. oxytoca*.

### Biosynthesis of the *Klebsiella* pyrazines

*Klebsiella* pyrazines **5-9** are previously unreported metabolites that are reminiscent of autoinducers produced by other bacteria. Specifically, compounds **5** and **7** contain a presumed aminoacetone (AA) moiety that is incorporated in established autoinducer compounds like DPO and AI-3[35,36]. Given the biological relevance of this compound class, we proceeded with targeted genetic analysis to establish the genetic basis of pyrazine biosynthesis in *K. oxytoca* and to establish mutants for biological comparisons.

We envisioned that pyrazines **5–9** could form through cyclization and oxidation events between the threonine metabolism product AA and γ-keto-δ-amino acid derivatives of phenylalanine (**12**) or tyrosine (**13**) (Fig. 3a). Structurally, **12** and **13** are analogs of 5-aminolevulinic acid (ALA), therefore we term them PALA and TALA for phenylalanine-

and tyrosine-derived ALA analogs, respectively. We hypothesized that compounds **12** and **13** could be formed through a PLP-dependent enzyme that decarboxylates and couples phenylalanine or tyrosine substrates with succinyl coenzyme A: an idea that is consistent with our finding that pyrazine formation relies on carbohydrate metabolism (Fig. 3a). Indeed, metabolite **12** was previously reported from *Streptomyces sp*. IB2014/O16-6, and its biosynthesis from phenylalanine and succinyl coenzyme A is mediated through the PLP-dependent 2-amino-3-ketobutyrate coenzyme A ligase PqrA[53]. In that study, **12** was an intermediate in the biosynthesis of a family of compounds termed perquinolines (Fig. S3a). We identified a PqrA homolog in *K. oxytoca*, which we term Pyr for pyrazine biosynthesis protein. The *K. oxytoca pyr* gene is part of an operon consisting of a tyrosine permease and a transporter protein (Fig. 3b). To assess if Pyr was involved in the biosynthesis of the *K. oxytoca* pyrazines, we generated a *pyr* mutant strain (Δ*pyr::spec*) and assessed metabolite production. Volcano plot analysis of the differentially regulated molecular features between the wildtype and Δ*pyr K. oxytoca* strains indicated that the pyrazines are dependent on the *pyr* gene (shown as over 200-fold upregulated in the wildtype via global XCMS[54] analysis) (Fig. 3c). Manual inspection of the extracted ion chromatograms of pyrazines **5–9** confirmed that production of these metabolites was completely abolished in the Δ*pyr* strain (Fig. S3b). These molecules could also be monitored by diode array detection at 310 nm due to their distinct chromophores. Doing so revealed striking differences in their 310 nm chromatograms which we confirmed arose from the presence of pyrazines in the wildtype cultures (Fig. 3d). Additionally, in agreement with the biosynthetic model, we were able to detect molecular features consistent with **12** and **13** which could not be detected in the Δ*pyr* strain (Fig. 3e). While the *pyr* gene is required for pyrazine production, there was little difference in yersiniabactin production (**10a**, **10b**) between the wildtype and Δ*pyr* strains (Fig. S3c). Since pyrazines **5–9** are derived from either homodimerization or heterodimerization among PALA, TALA, or AA substrates, we term them TALA-AA-Py (**5**), TALA-Py (**6**), PALA-AA-Py (**7**), PALA-TALA-Py (**8**), and PALA-Py (**9**).

To determine the role of threonine metabolism on pyrazine production and biosynthesis, we constructed a *tdh* mutant strain (Δ*tdh::spec*) and assessed relative pyrazine levels between the mutant and wildtype strains. We found that only production of pyrazines **5** and **7**, which incorporate *tdh*-derived aminoacetone, were significantly reduced in the Δ*tdh* strain, while the abundances of pyrazines **6, 8**, and **9** were not significantly different (Fig. S3d). These findings support that threonine metabolism only partially contributes to the pyrazine network via aminoacetone substrate supply in *K. oxytoca*.

To examine if the Pyr protein was sufficient for pyrazine production, we heterologously expressed the *pyr* gene in *E. coli* BL21(DE3) on a pET28a vector. In contrast to the empty vector controls, the Pyr expressing strain reconstituted pyrazines **5–9**, with compounds **6, 8**, and **9** being most prominent (Fig. 3f). Biochemical analysis of the substrate stereochemistry of the purified Pyr protein showed that it transformed either L- or D- phenylalanine or tyrosine, to **12** and **13**, respectively, but D-amino acid substrate utilization resulted in significantly increased levels of side products **14–17** (Fig. S4a). Compounds **14** and **16** were previously described from the enzymatic reaction of the *Streptomyces* Pyr-homolog as spontaneous *N*-succinylation products of phenylalanine and **12**, respectively[53]. Consistently, proposed metabolites **15** and **17** could be spontaneous *N*-succinylation products of tyrosine and **13**, respectively (Fig. S4b). Finally, through chemical synthesis of *R* and *S* enantiomers (Fig. S4c) coupled with Marfey's analysis[55], we were able to establish the stereochemistry of natural **12**, in culture, as being in the *S* configuration (Fig. S4d).

### Pyrazine and pyrazinone signaling regulates iron acquisition

We were intrigued by our observation that a specific sugar Neu5Ac, known to be derived from human mucosal degradation, regulates

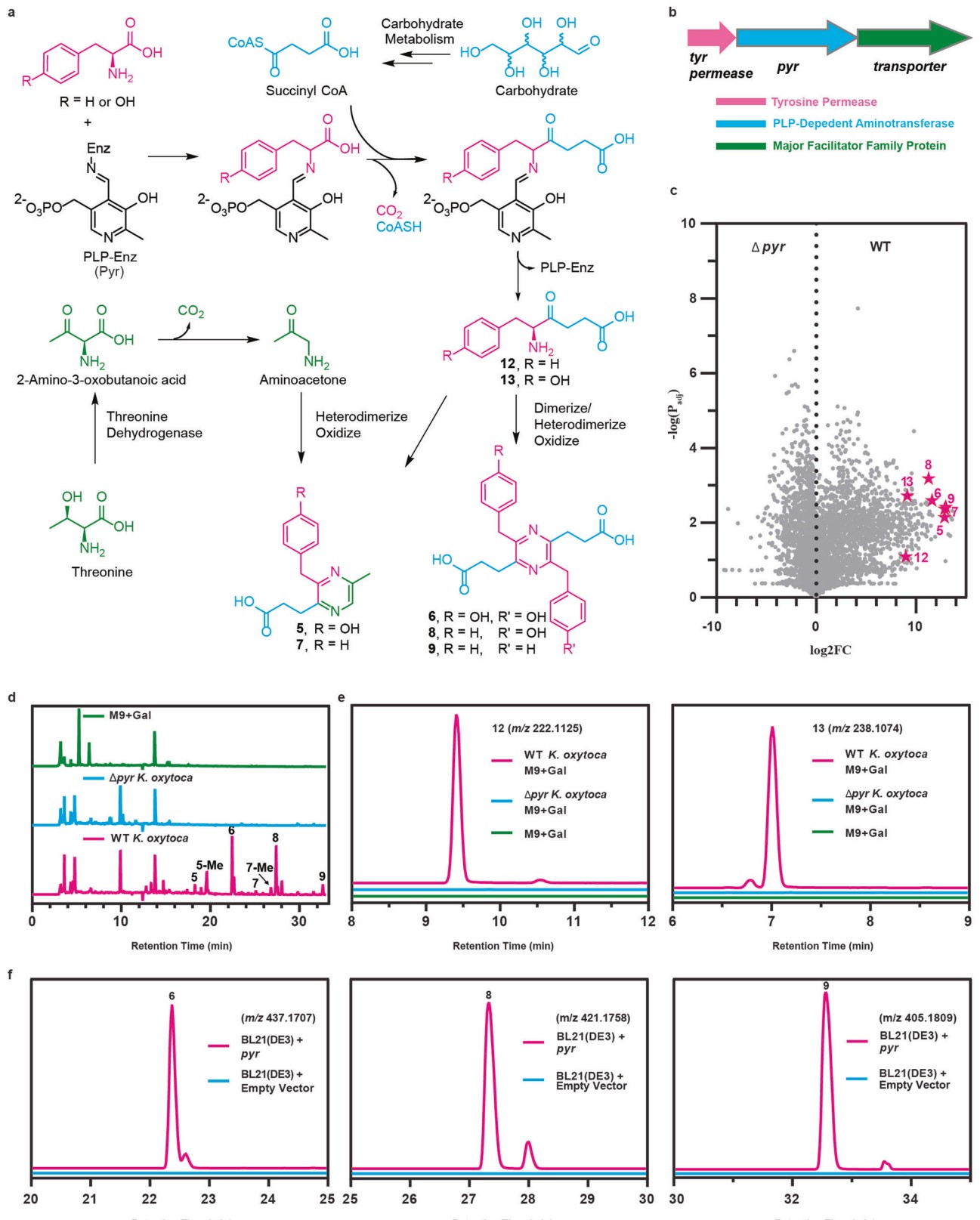

**Fig. 3 | Biosynthesis of *K. oxytoca* pyrazines. a** Proposed biosynthesis of *K. oxytoca* pyrazines. Molecular moieties are color coded to delineate chemical precursors. **b** Genetic architecture of the *pyr* operon in *K. oxytoca*. **c** XC-MS-based volcano plot analysis of chemical features differentially regulated between wildtype and Δ*pyr K. oxytoca* strains grown in M9 medium supplemented with D-galactose. Pyrazines **5**–**9** (TALA-AA-Py, TALA-Py, PALA-AA-Py, PALA-TALA-Py, and PALA-Py, respectively) and precursors **12** (PALA) and **13** (TALA) are marked. **d** 310 nm absorbance chromatograms of wildtype *K. oxytoca*, Δ*pyr K. oxytoca*, and medium control samples. **e** HPLC-QTOF-MS extracted ion chromatograms of **12** and **13** in wildtype *K. oxytoca*, Δ*pyr K. oxytoca*, and medium control samples. **f** HPLC-QTOF-MS extracted ion chromatograms of compounds **6**, **8**, and **9** from heterologous expression of the *K. oxytoca pyr* gene in *E. coli* BL21(DE3). For panels **d**, **e**, and **f**, representative traces of three biological replicates are shown ($n = 3$). For panel **c** statistical analysis was performed through a paired Welch t-test.

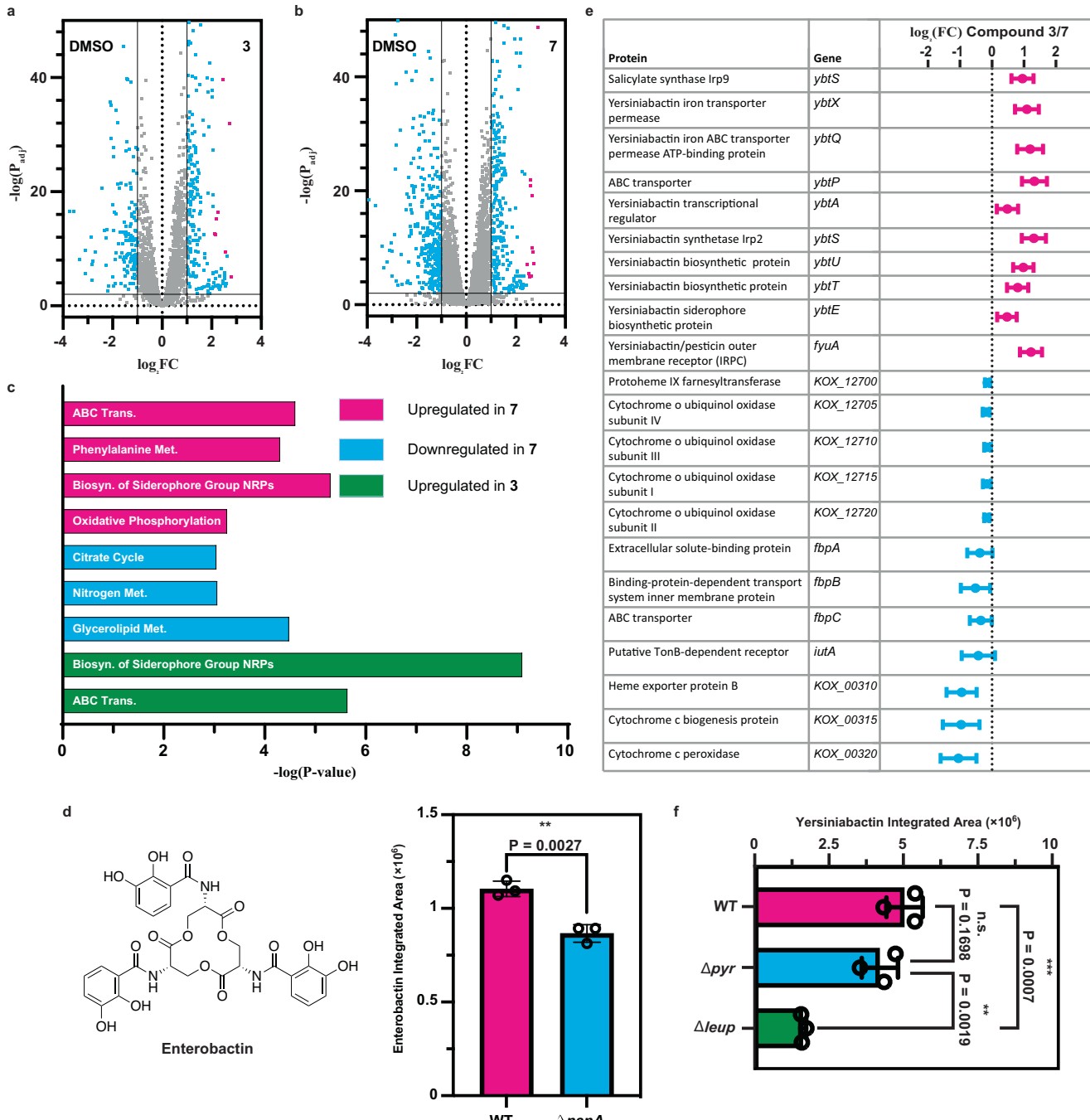

**Fig. 4 | Global effects of pyrazinone 3 and pyrazine 7 on *K. oxytoca* transcription. a** RNA-Seq volcano plot analysis of genes differentially regulated by pyrazinone **3** relative to a DMSO vehicle control. Points marked in blue have a $-\log(P_{adj}) > 2$ and abs($\log_2$FC) > 1. Points marked in red are related to iron acquisition pathways. **b** RNA-Seq volcano plot analysis of genes differentially regulated by pyrazine **7** relative to DMSO control. Points marked in red are related to iron acquisition pathways. **c** KEGG pathway analysis of pathways significantly regulated by **3** or **7**. No KEGG pathways were downregulated by the addition of **3** in a statistically significantly manner via the Funage-Pro web-based platform. **d** Structure of enterobactin and relative abundance of enterobactin between wildtype and Δ*nanA*

*K. oxytoca* strains supplemented with Neu5Ac (*n* = 3 biological replicates, *P* = 0.0027). **e** Overview of the most prominent genes differentially regulated between pyrazinone **3** and pyrazine **7**. **f** Relative abundance of yersiniabactin **10** between wildtype, Δ*pyr*, and Δ*leup K. oxytoca* strains (*n* = 3 biological replicates). Representative **10b** was selected. *P*-values between wildtype and Δ*pyr* strains, Δ*pyr* and Δ*leup* strains, and wildtype and Δ*leup* strains are 0.1698, 0.0019, and 0.007, respectively. Statistical analyses performed through an unpaired two-tailed t-test; n.s. indicates a non-significant difference, * indicates *p* < 0.05; ** indicates *p* < 0.01, and *** indicates *p* < 0.001. Data are represented as mean values +/− SD.

both leupeptin/pyrazinone and aminoketone/pyrazine production, whereas diverse sugars that may be available in luminal environments only support aminoketone/pyrazine production. As such, we endeavored to better understand the signaling responses of pyrazines versus pyrazinones in *K. oxytoca* via RNA-sequencing experiments. We used the Δ*pyr K. oxytoca* background under non-

*leup* inducing conditions (D-Gal supplementation), cultured this "non-producing" strain to mid-exponential growth, and then supplemented with pyrazinone **3** (10 μM), pyrazine **7** (10 μM), or DMSO vehicle control. Volcano plot analysis of the differentially regulated genes in both the pyrazinone **3**-supplemented (Fig. 4a) or pyrazine **7**-supplemented (Fig. 4b) cultures, relative to the DMSO control,

revealed that compound supplementation had profound effects on the *K. oxytoca* transcriptome. Specifically, pyrazinone **3**-supplementation differentially regulated 561 genes ($P_{adj}$ < 0.05; abs(log$_2$FC) > 1) while pyrazine **7**-supplementation differentially regulated 644 genes ($P_{adj}$ < 0.05; abs(log$_2$FC) > 1). Notably, inspection of the genes most prominently upregulated by either **3** or **7** indicated that these compounds modulated the transcription of genes related to iron acquisition and iron metabolism, marked in red in Fig. 4a and Fig. 4b. Similarly, examining the role of pyrazine **6** (no AA incorporation) on *K. oxytoca* found that supplementation of this compound also differentially regulated the transcriptome (201 genes with $P_{adj}$ < 0.05; abs(log$_2$FC) > 1) and enhanced the transcription of genes related to iron acquisition and iron metabolism (Supplemental Information).

A systematic analysis of the pathways regulated by pyrazinone **3** and pyrazine **7** through the Funage-Pro online platform[56] revealed that both compounds upregulated pathways associated with the biosynthesis of siderophore nonribosomal peptides (NRPs) (Fig. 4c). This finding is consistent with the upregulation of iron acquisition genes, given the established role of NRP siderophores in iron uptake[57]. Furthermore, pyrazine **7** also upregulated pathways involved in ATP-binding cassette (ABC) transport, phenylalanine metabolism, and oxidative phosphorylation. Conversely, pathways associated with nitrogen metabolism, glycerolipid metabolism, and the citric acid cycle were downregulated by **7**. Similar to **7**, pyrazinone **3** upregulated pathways involved with ABC transport, though no conserved pathways were significantly downregulated in this analysis platform.

Examining the RNA-sequencing data for siderophore production uncovered that both **3** and **7** significantly enhanced transcription of the enterobactin BGC (Fig. 4d). Notably, this response was not unique to pyrazine **7** as pyrazine **6** also exhibited increased enterobactin transcription levels relative to a DMSO control (Supplemental Information). Enterobactin boasts a remarkably high affinity for ferric iron[58] and is one of the most commonly used siderophores in Gram-negative bacteria[59]. To validate the effect of pyrazine and pyrazinone signaling on enterobactin biosynthesis, we compared wildtype and Δ*nanA K. oxytoca* cultures supplemented with Neu5Ac (leupeptin and pyrazine stimulating conditions) by LC-MS. Indeed, we found a statistically significant increase in enterobactin production in the wildtype relative to the Δ*nanA* strain (Fig. 4d). A similar trend was observed for the monomer, dimer, and linear trimer of enterobactin (Supplemental Information). These findings imply that both *K. oxytoca* pyrazines and pyrazinones can promote enterobactin production and iron metabolism.

Next, we sought to discern signaling differences between pyrazinone **3** and pyrazine **7**. An examination of BGCs differentially regulated by these compounds revealed that pyrazinone **3** prominently upregulated the yersiniabactin BGC, while cytochrome-related operons were prominently upregulated by pyrazine **7** (Fig. 4e). Several of the cytochrome operons are linked to heme-iron chemistry and electron acceptors in aerobic respiration, a finding consistent with **7**'s upregulation of pathways associated with oxidative phosphorylation. Intriguingly, the finding that **3** upregulates the established *Klebsiella* virulence factor yersiniabactin[60] relative to **7** implies that pyrazinone signaling may be involved in pathogenesis.

To validate that pyrazinone **3** affects the production of yersiniabactin (**10**), we cultivated wildtype, Δ*pyr*, and Δ*leup K. oxytoca* with Neu5Ac supplementation and employed LC-MS to assess the extracts for the presence of **10** (Fig. 4f). Under these conditions, we found that the wildtype strain produced the most **10**, production was statistically unaffected in Δ*pyr*, and Δ*leup* showed a significant reduction relative to wildtype. These findings align with the hypothesis that leupeptin-derived pyrazinones contribute to the regulation of yersiniabactin.

## Pyrazinone 3 selectively activates histamine receptor H4

Due to our discovery that pyrazinone **3** has the capacity to promote virulence factor biosynthesis in *K. oxytoca*, we were interested in exploring other mechanisms by which **3** could interact with human cells. Thus, we screened **3** against a 314-member panel of human G protein-coupled receptors (GPCRs) via PRESTO-Tango analysis (Fig. 5a)[37]. Additionally, we also subjected a homoarginine derivative of **3** (**3**-**Homo**), which is produced by *Xenorhabdus* and *Photorhabdus leup* operons, but not *K. oxytoca*, to the same analysis[12]. Intriguingly, both **3** and **3**-**Homo** emerged as selective agonists for histamine receptor H4 (HRH4), with minimal activity towards other GPCRs in the panel. The histamine receptor family encompasses four receptors (HRH1, HRH2, HRH3, and HRH4), which collectively regulate a plethora of processes including allergic inflammation[61], immunomodulation[62], gastric acid secretion[63], and neurotransmission[64,65]. HRH4, in particular, is expressed by several cells of the immune system, including monocytes, eosinophils, dendritic cells, T cells, and natural killer cells, and is known to regulate immune functions[66]. Interestingly, activation of HRH4 is associated with exacerbated respiratory dysfunction[66]. Notably, the finding that the *leup* product **3** is an agonist of HRH4 is consistent with the prevalence of the leupeptin BGC in *K. oxytoca* associated with hospital-acquired respiratory infections.

Next, we tested the affinities of pyrazines **6, 7**, and **9**-**Me** and pyrazinones **3** and **3**-**Homo** towards the individual histamine receptors, utilizing the canonical substrate histamine as a control (Fig. 5b). Histamine activated HRH1, HRH2, HRH3, and HRH4 with EC$_{50}$ values of 4.16 nM, 0.22 mM, 283 nM, and 419 nM, respectively. Interestingly, pyrazinones **3** and **3**-**Homo** exhibited EC$_{50}$ values of 12.5 μM and 2.7 μM, respectively, towards HRH4, whereas the pyrazines displayed no activity towards any of the histamine receptors. These findings support the proposal that pyrazinone signaling in *K. oxytoca* may exacerbate proinflammatory signaling via HRH4 GPCR activation in the lung environment.

## *Klebsiella* pyrazines are broadly conserved across bacteria

Considering the significant role of the pyrazines in regulating iron acquisition and metabolism in *K. oxytoca*, we endeavored to identify other bacteria that encode related compounds. Thus, we performed a protein BLAST search using the IMG database to find homologs of the *K. oxytoca* Pyr protein and assembled top hits with an E-score lower than 0.05 into a phylogenetic tree (Fig. 6a). The phylogenetic tree highlights that pyrazine producing machinery is conserved across diverse bacterial families. The *pyr* gene appears either as a standalone gene or as part of divergent operon structures, suggesting that the Pyr protein, in addition to being necessary and sufficient for pyrazine biosynthesis, may participate in building core precursors for other specialized metabolic pathways. This notion is supported by work in *Streptomyces sp*. IB2014/016-6, which demonstrated that the Pyr-homolog, PqrA, facilitates the production of **12** en route to the perquinolines[53]. Notably, the *leup* operon is more narrowly distributed than the *pyr* gene[12].

To glean functional insights into Pyr proteins across bacterial families, we created a sequence similarity network (SSN)[67] comprising all the Pyr proteins identified in our BLAST analysis (Fig. 6b). When categorized by percent identity, the nodes of the SSN clustered along bacterial taxonomies. Interestingly, some taxonomic families, like *Pseudomonadaceae*, exhibited several versions of the Pyr protein that clustered separately, suggesting broader structural diversification of Pyr-associated metabolites. The *Klebsiella* Pyr protein clustered closely with Pyr-homologs from Shiga toxin-producing *E. coli* (STEC) and *Photorhabdus*, suggesting that these bacteria may also produce related metabolites.

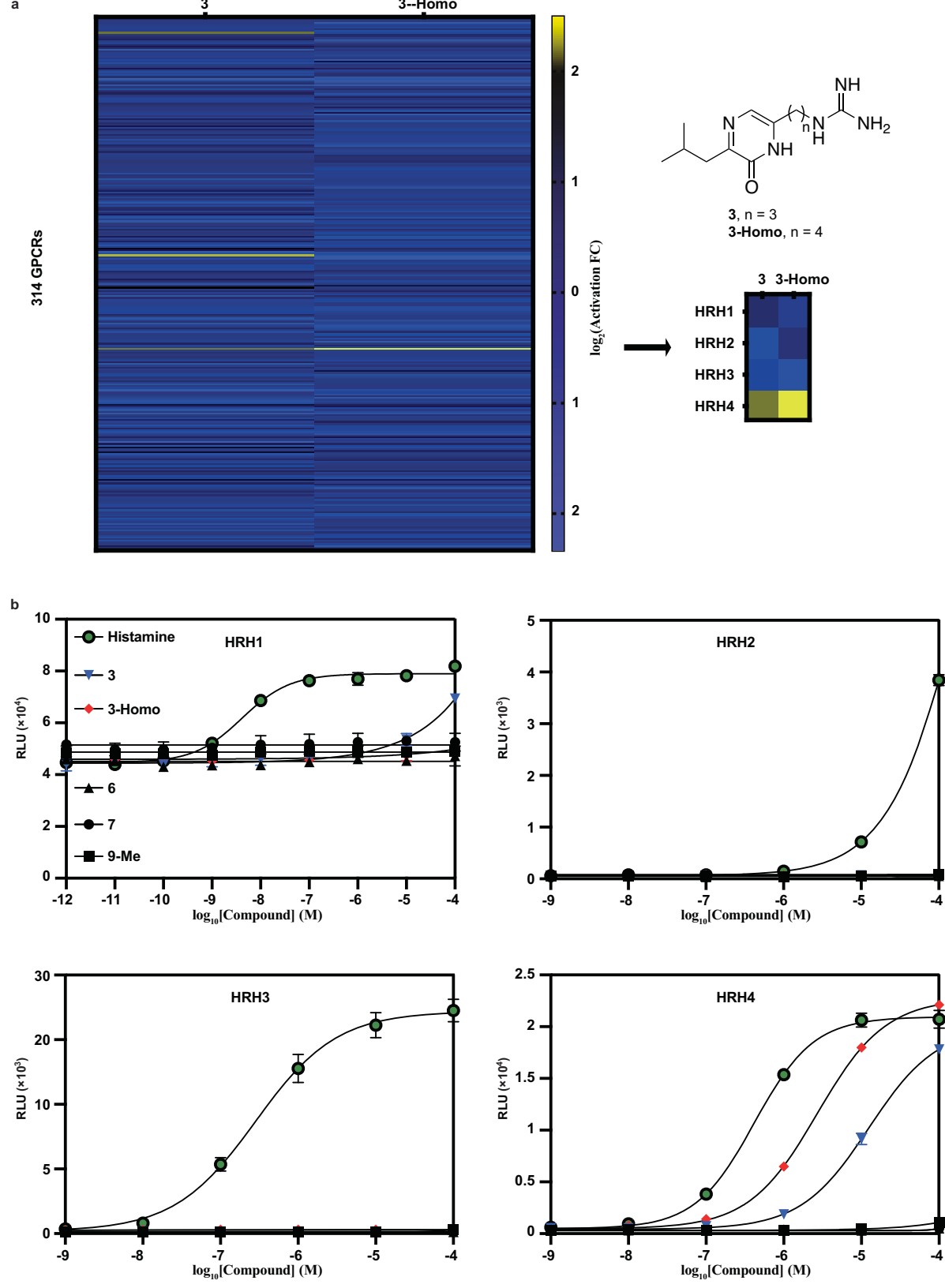

**Fig. 5 | Affinity of pyrazines and pyrazinones towards human GPCRs. a** Heatmap depicting the activation of 314 human GPCRs by compounds **3** and **3-Homo** (100 μM) as analyzed by PRESTO-Tango. The right image illustrates a zoomed in region of the heatmap centered around the histamine receptors. **b** Dose-response curves of compounds **3, 3-Homo, 6, 7, 9-Me**, and histamine against the GPCRs HRH1, HRH2, HRH3, and HRH4 ($n = 3$ biological replicates). Luminescence was measured in relative luminescence units (RLUs). Data are represented as mean values +/− SD.

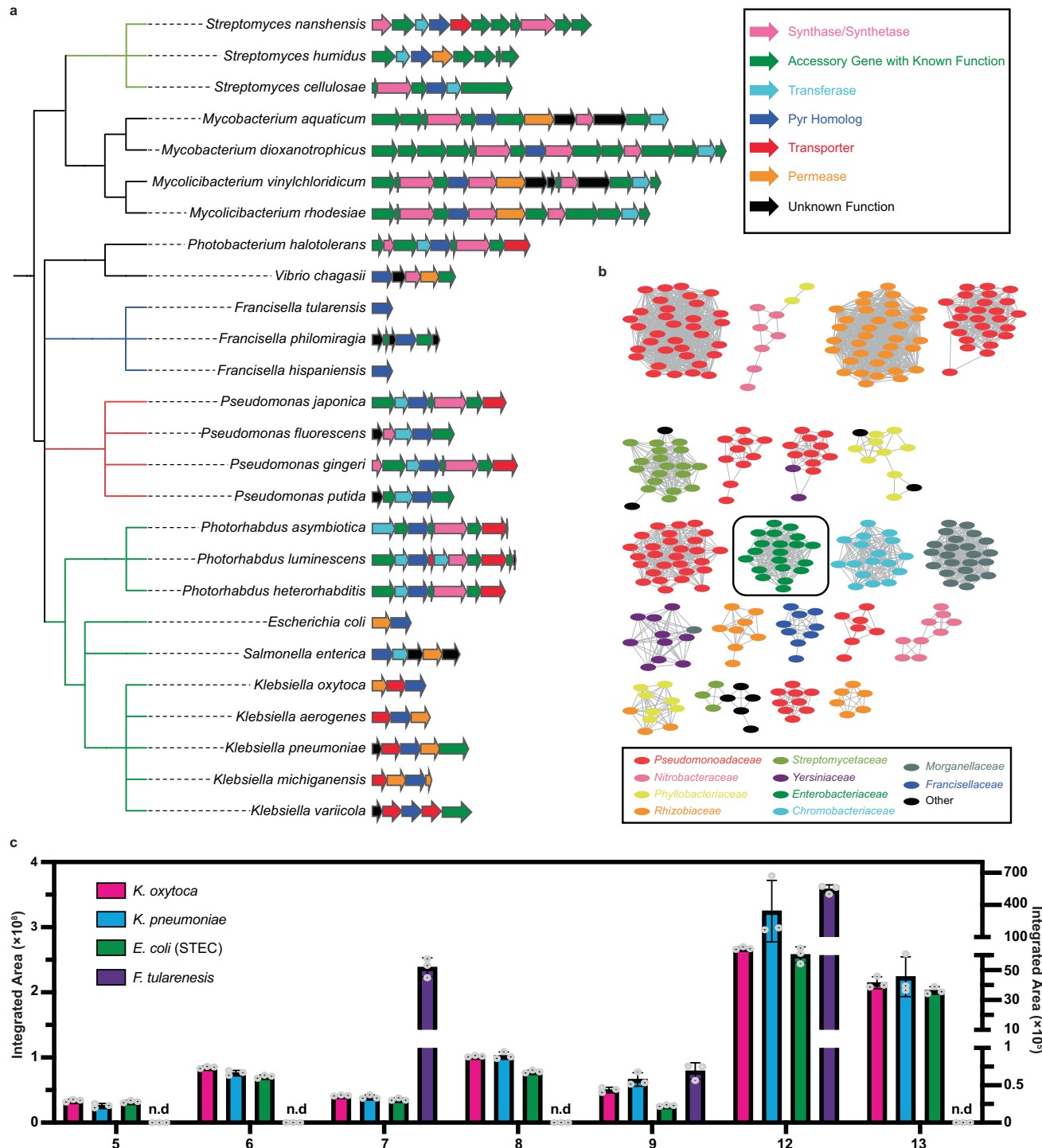

**Fig. 6 | Distribution of pyrazine biosynthetic machinery across bacteria.**
**a** Phylogenetic analysis of Pyr homolog-containing operons across various families of bacteria. Bacterial families are colored as follows: green, Enterobacteriaceae; red, Pseudomonoadaceae; blue, Francisellaceae; olive, Streptomycetaceae; and black, other. Functional annotations of the genes are separately color coded and displayed in the top-right insert. **b** Protein similarity networking analysis of the Pyr protein across various families of bacteria. Connected nodes indicate high protein similarity (>90% sequence homology). Bacterial families are color coded and displayed in the bottom-right insert. The Pyr network that contains *Klebsiella* species (in the Enterobacteriaceae family) is boxed. **c** Production of pyrazines **5**–**9** and monomers **12** and **13** by select Pyr homologs in the heterologous host *E. coli* BL21(DE3). The left axis plots *K. oxytoca*, *K. pneumoniae*, and *E. coli* (STEC) integrated areas, while the right axis plots the *F. tularensis* integrated areas. Data are represented as mean values +/− SD.

To confirm that pyrazine production is indeed conserved across bacterial families, we heterologously expressed Pyr homologs from *Klebsiella pneumoniae* (Kp589), *Francisella tularensis* (WY96), and STEC (3055), and assessed the production of emergent metabolites. Heterologous expression of the Pyr proteins from these selected pathogens demonstrated that all of them were capable of producing pyrazines (Fig. 6c). *K. pneumoniae* and STEC Pyr proteins produced pyrazines **5**-**9** and their monomers **12** and **13** in comparable levels to that of the *K. oxytoca* Pyr protein, with only pyrazine **9** showing a noticeable decrease in production in STEC. Meanwhile, the *F.*

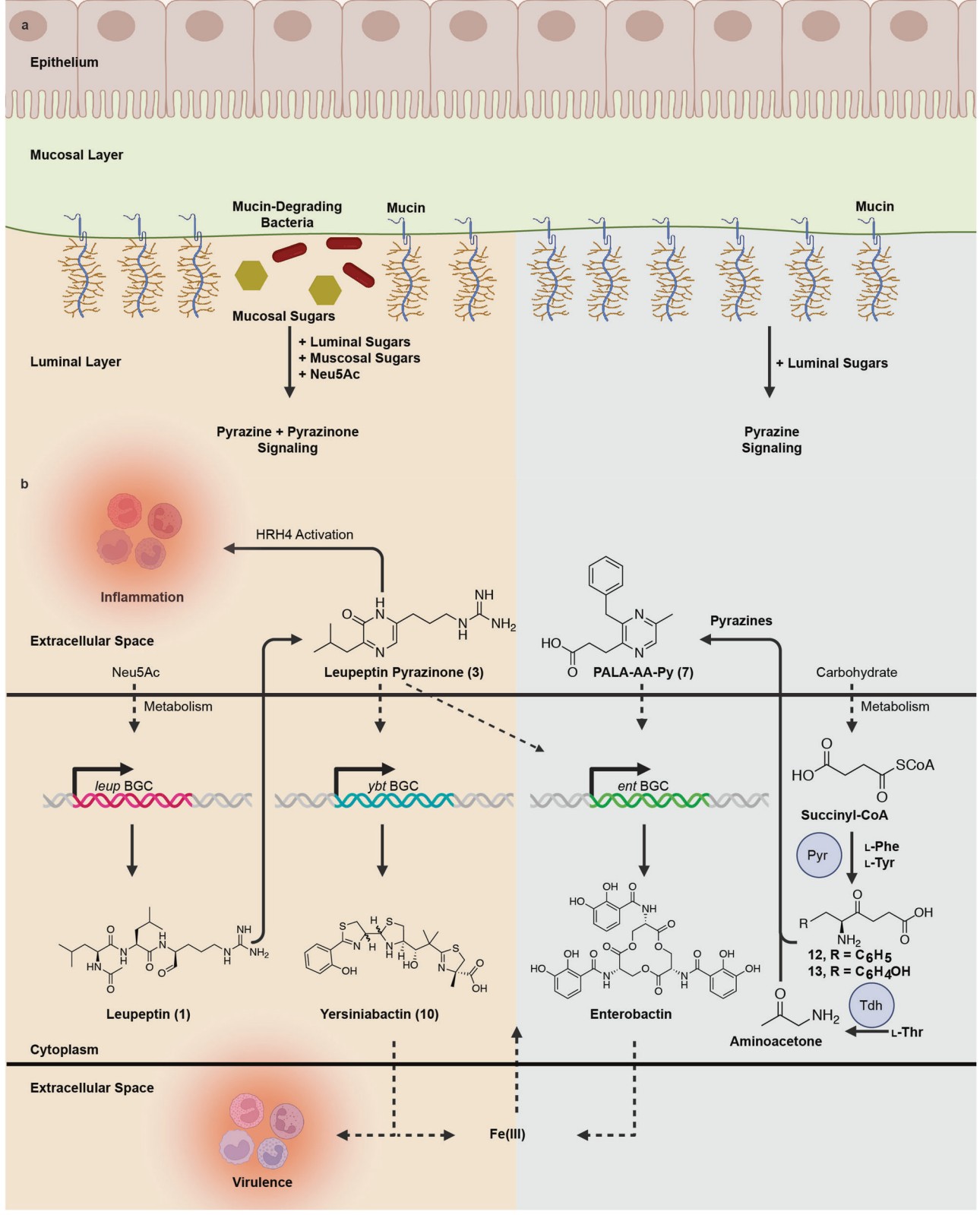

*tularensis* Pyr could only produce pyrazines **7** and **9** and monomer **12**, suggesting some level of pyrazine selectivity across taxa. The genetic distribution of the *pyr* gene and confirmation that *pyr*-encoding bacteria beyond *K. oxytoca* produce pyrazines **5**-**9** supports that pyrazine biosynthesis is ubiquitous and may serve diverse biological roles in chemical signaling. Notably, while Pyr-derived pyrazines regulate iron acquisition systems in *K. oxytoca*, additional studies are required to determine their broader biological functions across bacterial lineages.

## Discussion

In this work, we highlight the identification, biosynthesis, and chemical signaling responses of differentially-regulated pyrazine and pyrazinone autoinducer systems in *K. oxytoca* (Fig. 7). While natural

**Fig. 7 | Proposed model for differential pyrazine versus pyrazinone signaling in *K. oxytoca* human lung isolates. a** Proposed mechanism of pyrazine versus pyrazinone induction based on carbohydrate utilization. In the presence of mucin-degrading bacteria (left, light-orange section), mucosal sugars—specifically Neu5Ac—and luminal sugars are released into the environment. The metabolism of Neu5Ac leads to *leup*-derived pyrazinone signaling and *pyr*-derived pyrazine signaling. Notably, pyrazine signaling is not restricted to Neu5Ac metabolism but relies on carbohydrate metabolism more broadly. In the absence of mucin-degrading bacteria (right, light-blue section), *K. oxytoca* metabolizes luminal sugars, resulting in *pyr*-derived pyrazine signaling. **b** Overview of pathways differentially impacted by pyrazine versus pyrazinone signaling in *K. oxytoca*. When *K. oxytoca* metabolizes Neu5Ac, the *leup* operon is activated leading to the production of leupeptin **1** and its pyrazinone derivative **3** (left, light-orange section). Leupeptin pyrazinone **3**

activates human HRH4, potentially causing an inflammatory response, and enhances production of the siderophores yersiniabactin **10** and enterobactin, which scavenge for environmental iron and other metals. Yersiniabactin, specifically, serves as a host virulence factor in *Klebsiella*. Under broad carbohydrate metabolism conditions, including Neu5Ac, Pyr (pyrazine biosynthesis enzyme) couples L-Phe or L-Tyr with succinyl-CoA to form PALA **12** and TALA **13** (right, light-blue section). These compounds can dimerize with themselves, each other, or AA to form pyrazines **5–9**. While all of these compounds may affect the *K. oxytoca* transcriptome, RNA-sequencing shows that representative PALA-AA-Py **7** stimulates enterobactin biosynthesis, promoting iron acquisition. The *leup* BGC, *ybt* BGC, and *ent* BGC refer to the leupeptin, yersiniabactin, and enterobactin biosynthetic gene clusters, respectively. Figure was created with BioRender.com[82].

products containing pyrazinone scaffolds, such as DPO and AI-3, have emerged as potent regulators of bacterial quorum sensing phenotypes, like biofilm formation and virulence factor production in human-associated bacteria, pyrazine secondary metabolites have been comparably less characterized from the human microbiome. Pyrazines have been discovered in environmental bacteria such as *Xenorhabdus indica*[68], *Bacillus subtilis*[69], *Serratia marcescens*[70], and *Pseudomonas fluorescens*[71], though the roles of these compounds in bacterial signaling is unknown. Recently, *Pseudomonas* bacteria have been shown to produce pyrazine *N*-oxides which regulate the production of virulence factors and are thought to contribute to host colonization[72–74]. With respect to this study, the *K. oxytoca* pyrazines have been shown to regulate the transcription of metabolism and iron acquisition systems and the genes responsible for their synthesis are conserved among diverse bacteria. Regulation of iron acquisition systems was also observed in a study on the global transcriptomic response of DPO and AI-3 in *S. enterica* Typhimurium[75]. Interestingly, that study found that supplementation of DPO and AI-3 at supra-physiological levels (100 μM) downregulated iron acquisition pathways including enterobactin biosynthesis and uptake[75]. This result contrasts with our findings that the *K. oxytoca* pyrazinone **3** and pyrazine **7** upregulate iron acquisition pathways and may allude to either concentration-dependent effects or differential bioactivities of pyrazinone autoinducers across bacterial taxa.

This work also found that the human mucin-capping sugar, Neu5Ac, served as the elicitation signal for the *K. oxytoca* leupeptin BGC: an operon that has been correlated to nosocomial respiratory infections[12]. We were intrigued by Neu5Ac's role as a selective elicitor of leupeptin/pyrazinone production relative to pyrazine biosynthesis, which is regulated by general carbohydrate metabolism. This finding may support a model in which *K. oxytoca* utilizes pyrazine-based chemical signaling for competition in diverse carbohydrate environments, such as the lumen, and dual pyrazine- and pyrazinone-based chemical signaling for competition at the mucosal barrier during mucosal degradation in environments such as the lung. Notably, carbohydrate metabolism in *K. oxytoca* has been linked to in vivo colonization resistance against pathogenic bacteria like *K. pneumoniae*[6]. While this phenotype is dependent on the ability of *K. oxytoca* to utilize a broader panel of sugars relative to *K. pneumoniae*[6], our findings also suggest that sugar-dependent changes are accompanied with significant downstream chemical signaling responses. Indeed, while this manuscript was in review, a paper was published detailing how tilimycin production, both in vitro and in vivo, could be linked to the metabolism of simple carbohydrates[76]. The *K. oxytoca* strain used in our study lacks the tilimycin pathway, so potential pyrazine/pyrazinone effects on tilimycin enterotoxin production remain unresolved.

Due to the *leup* BGC's association with respiratory infections, its activation and the differential transcriptional responses this causes in *K. oxytoca* may be of clinical importance. In this study, we found that activation of the *leup* BGC leads to the production of leupeptin-derived

pyrazinones, as previously described in *Xenorhabdus bovienii*[12]. Through PRESTO-Tango-based GPCR analysis, we found that pyrazinone **3** and its homoarginine derivative, **3-Homo**, are selective agonists for the human GPCR HRH4. HRH4 is expressed in the lung[77] where it has been shown to mediate allergic airway inflammation by influencing CD4⁺ T cell activation[78]. Thus, nosocomial respiratory infections associated with *leup*⁺ *K. oxytoca* may be mechanistically linked to production of pyrazinone **3** and subsequent activation of HRH4 in addition to the broad spectrum protease inhibition of the leupeptin precursors. Furthermore, we found that production of pyrazinone **3** has transcriptional consequences on the *K. oxytoca* secondary metabolome, namely increased production of yersiniabactin **10**. Yersiniabactin is a well-characterized virulence factor which has been linked to respiratory tract infections[79,80]. In *K. pneumoniae*, yersiniabactin is required for maximal growth and lethality during pulmonary infection and promotes respiratory tract infection through evasion of host lipocalin 2[80]. Altogether, our findings suggest that the pathogenicity of *K. oxytoca* strains in the human lung may be related to *leup*-mediated chemical signaling in both the bacteria and the host.

## Methods

### General experimental parameters and instrumentation
High-resolution electrospray ionization mass spectrometry (HRMS) was performed on an Agilent iFunnel 6550 quadrupole time-of-flight (QTOF) mass spectrometry instrument coupled to an Agilent 1290 Infinity HPLC system and a Kintex 5 μm $C_{18}$ 100 Å column (250 × 4.6 mm). Chromatography was performed with a linear water-acetonitrile gradient containing 0.1% formic acid at 0.7 mL/min over 30 min. The gradient proceeded from 5% acetonitrile to 50% acetonitrile, followed by either a 3 or 10-minute 100% acetonitrile wash. Mass spectra were recorded in positive ionization mode with a mass range of 100 $m/z$ to 1700 $m/z$. Targeted $MS^2$ analysis was performed with Iso width set to 1.3 $m/z$ (narrow width) and a fixed collision energy of 20 CE. UV-vis spectra were obtained on an Agilent 1260 Infinity system equipped with a photo diode array (PDA) detector. NMR spectroscopy was recorded on either an Agilent 600 MHz NMR spectrometer (DD2) equipped with an inverse cold probe (3 mm), employing standard NMR pulse sequences, or a Bruker 400 MHz NMR spectrometer equipped with a broadband probe, also employing standard NMR pulse sequences.

### Genetic recombineering in *K. oxytoca*
*K. oxytoca* ATCC 8724 was transformed with pORTMAGE-3 (Addgene 72678)[38]. Briefly, 1 mL of mid-log *K. oxytoca* in LB was pelleted, washed twice with ice cold water and concentrated to 50 μL. 50 ng of pORTMAGE-3 was added to the cells and the concentrate was loaded into a 1 mm electro cuvette and pulsed at 1800 V, 25 μF, 200 Ω (BTX Gemini X2). Cells recovered overnight at 30 °C in LB broth and were plated on LB agar plates supplemented with kanamycin (50 μg/mL) at 30 °C.

For generation of the *K. oxytoca* leupeptin reporter strain, PCR was used to generate a GFP-spectinomycin oligonucleotide fusion with 45 bp homologous arms directly upstream and downstream of the *leupA* gene. A list of all primers used in this study are included in the Supplemental Information. 50 ng of this PCR product was electroporated into activated *K. oxytoca* containing pORTMAGE-3. Cells recovered overnight at 30 °C in LB broth and were plated on LB agar plates supplemented with spectinomycin (100 μg/mL) at 30 °C. The reporter strain was cured of pORTMAGE-3 by culturing in LB + spectinomycin at 30 °C and back diluting cultures 1:1000 until the strain lost its kanamycin resistance.

For generation of the Δ*nanA*, Δ*pyr*, and Δ*leup K. oxytoca* mutants, PCR was used to generate spectinomycin cassette oligonucleotides with 45 bp homologous arms directly upstream and downstream of the genes (or pathway in the case of Δ*leup*) to be deleted. Lambda red recombineering and plasmid curing was performed as described above. The sequences of all primers used in this study are including in the Supplementary Information.

## Leupeptin detection assays

5 mL of the leupeptin reporter strain was cultured aerobically in 10 g/L of LB (Becton Dickinson), yeast extract, GAM (HiMedia Laboratories), horse liver extract (HiMedia Laboratories), malt extract (VWR), meat extract (HiMedia Laboratories), YPD (Dot Scienticifc), brain heart infusion (Becton Dickinson), porcine mucin (Sigma-Aldrich, M1778), clostridial (Becton Dickinson), TSB (VWR), or casamino acid supplemented (5 g/L, VWR) M9 minimal salts (MP Biomedicals) at 30 °C for 24 h at 250 rpm. Cultures were lysed via sonication, centrifuged, and 150 μL of the supernatants were aliquoted into black well clear bottom 96 well plates. GFP fluorescence measurements were performed on a BioTek Synergy H1 plate reader with an excitation wavelength of 488 nm and an emission wavelength of 525 nm. For the carbohydrate screen, the leupeptin reporter strain was cultured in M9 minimal medium (MPbio) supplemented with casamino acid (5 g/L, VWR) and 0.4% indicated sugar aerobically at 30 °C for 24 h at 250 rpm.

## Metabolomics analysis of *K. oxytoca* strains

For MS-based determination of leupeptin activation, wildtype *K. oxytoca* ATCC 8724 was cultured in 5 mL of M9 minimal medium supplemented with 5 g/L casamino acids and 0.4% Neu5Ac. A similar culture lacking Neu5Ac supplementation was also generated as a comparison. The cultures were incubated at 30 °C for 48 h at 250 rpm and then dried at room temperature under reduced pressure (Genevac). The dried samples were resuspended in 200 μL of 1:1 methanol:water, and 3 μL of the samples were subjected to MS and $MS^2$ profiling via HPLC-QTOF-MS (with PDA detector), as described above. The XCMS Online web-based platform[54] was used to identify molecular features present in the Neu5Ac supplemented cultures. Furthermore, the GNPS online repository[48] was used to link features with similar $MS^2$ spectra in a molecular network. This chromatography and MS protocol was used for all MS experiments in this work. When sugars other than Neu5Ac were used to supplement cultures, they were present at 0.4%. To measure the area under the curves for specific molecular features, the exact monoprotonated $[M + H]^+$ mass was calculated and extracted ion chromatographs (EICs) with a 10-ppm error were constructed. The peaks in the EICs were integrated with the Agilent Mass Hunter Qualitative Analysis software. UV-vis chromatographs were generated with a detection wavelength of 310 nm (bandwidth 4 nm).

## GNPS networking details

Molecular networks were created using the online workflow (https://ccms-ucsd.github.io/GNPSDocumentation/) on the GNPS website (http://gnps.ucsd.edu). The data were filtered by removing all $MS^2$ fragment ions within +/− 17 Da of the precursor *m/z*. $MS^2$ spectra were window-filtered by choosing only the top 6 fragment ions in the +/− 50 Da window throughout the spectrum. The precursor ion mass tolerance was set to 0.05 Da and a $MS^2$ fragment ion tolerance of 0.1 Da. A network was then created where edges were filtered to have a cosine score above 0.7 and more than 2 matched peaks. Further, edges between two nodes were kept in the network if and only if each of the nodes appeared in each other's respective top 10 most similar nodes. Finally, the maximum size of a molecular family was set to 100, and the lowest scoring edges were removed from molecular families until the molecular family size was below this threshold. The spectra in the network were then searched against GNPS spectral libraries. The library spectra were filtered in the same manner as the input data. All matches kept between network spectra and library spectra were required to have a score above 0.7 and at least 2 matched peaks. Nodes were annotated with the high-resolution masses of the molecular features fragmented in the metabolomic profiling experiment.

## Purification and structural elucidation of pyrazines

5 L of *K. oxytoca* ATCC 8724 was cultivated in M9 minimal medium supplemented with casamino acids (5 g/L) and 0.4% D-Gal at 30 °C and 250 rpm for 48 h. The culture was centrifuged at 5000 g for 30 min at room temperature, and the clarified supernatant was incubated with 10 g/L dry XAD-7 HP resin for 2 h at 37 °C and 150 rpm. The resins were pooled and extracted with 10 L of methanol which was subsequently dried at room temperature under nitrogen. The dried methanolic extract was resuspended in methanol (100 mL) and subjected to gravity column fractionation with LiChroprep $RP_{18}$ (500 g; 5 × 20 cm) with a step-gradient elution (0–100% methanol in water, 20% methanol increments, 500 mL each) to generate 6 fractions. Among these fractions, the 80% methanol fraction contained almost all of the compounds of interest, as determined by UV-Vis spectroscopy. This fraction was further purified by preparative scale RP-HPLC equipped with an Agilent Polaris $C_{18}$-A column (5–50% acetonitrile in water with 0.01% TFA for 60 min, 8 mL/min, 1 min collection interval). A subsequent purification of the fractions containing UV-vis signatures consistent with the compounds of interest via semi-preparative RP-HPLC equipped with a Phenomenex Luna $C_{18}$ column (5 to 50% acetonitrile in water with 0.01% TFA for 60 min, 4 mL/min, 1 min collection interval) led to the purification of compounds **6** (1.3 mg), **7** (3.0 mg), and **9-Me** (0.7 mg).

## Purification of *K. oxytoca* Pyr protein

A His-Tagged (C-terminal) variant of the *K. oxytoca* Pyr protein, optimized for *E. coli* expression, was ordered from Twist Bioscience on a pET28a vector and transformed into *E. coli* BL21(DE3). The coding sequence of Pyr-His$_{6x}$ is available in the SI. This strain was grown aerobically in 10 mL of LB medium with kanamycin selection (50 μg/mL) overnight at 37 °C at 250 rpm. The overnight culture was used to inoculate a 1 L culture of LB with kanamycin which was cultured under identical conditions until an $OD_{600}$ of 0.6. At this point, 100 μM final concentration of IPTG was added to the culture, and the samples were further incubated at 16 °C at 250 rpm for 16 h. Cultures were centrifuged at 5000 g for 30 min at 4 °C, and the pelleted cells were resuspended in 25 mL of Ni-NTA equilibration buffer (20 mM $NaH_2PO_4$, 300 mM NaCl, 10 mM imidazole, pH 7.4) with 1 cOmplete EDTA-free protease inhibitor tablet (Sigma-Aldrich). The suspension was sonicated on ice for 15 min (10 seconds on, 50 seconds off at 60% amplitude for 2.5 min) and then centrifuged at 19,000 g for 30 min at 4 °C. The lysate was loaded onto a column containing 2 mL of Ni-NTA resin which was equilibrated with 10 mL of Ni-NTA equilibration buffer. The column was washed twice with 4 mL of Ni-NTA wash buffer (20 mM $NaH_2PO_4$, 300 mM NaCl, 25 mM imidazole, pH 7.4) and then eluted with 2 mL of Ni-NTA elution buffer (20 mM $NaH_2PO_4$, 300 mM NaCl, 250 mM imidazole, pH 7.4). The elution was concentrated in a 30 kD centrifugation filter and buffer exchanged into PBS. Protein concentrations were measured by Bicinchoninic acid assay.

### In vitro biochemical reactions

Pyr biochemical reactions were performed in PBS buffer (pH 7.4) with an enzyme concentration of 25 μM, a supplemented PLP concentration of 80 μM, a succinyl-CoA concentration of 1 mM, and an amino acid (L-Phe, D-Phe, L-Tyr, D-Tyr) concentration of 2 mM. For each, 50 μL of reaction was incubated for 1 hour at room temperature and then quenched with 50 μL of acetonitrile. The reactions were then centrifuged for 5 min at 18,000 g, and 3 μL of the supernatants were subjected to HPLC-QTOF-MS analysis, as described above.

### Chemical synthesis of *S* and *R* 12

100 mg of either L- or D- *N*-Boc phenylalanine (Sigma-Aldrich) was incubated, with stirring, with 1.1 equivalents of carbonyldiimidazole in 3 mL of dry dichloromethane for 90 min at room temperature. At this point, 1.1 equivalents of *N,O*-dimethylhydroxylamine was added, and the solution was stirred overnight at room temperature to yield Weinreb amide **18**, which was isolated by HPLC purification (~86% yield). 5 equivalents of a 3-butenylmagnesium bromide dry tetrahydrofuran solution at 0 °C was added dropwise to 40 mg of **18** in 2 mL dry tetrahydrofuran. Once completely added, the solution was allowed to warm to room temperature and the solution was stirred for 4 h. The solution was quenched by slow addition of aqueous ammonium chloride and product **19** was purified by HPLC (~51% yield). 10 mg of **19** was dissolved in 2 mL of acetone. To this solution, 0.03 equivalents of aqueous $RuCl_3 \cdot 3H_2O$ and 5.5 equivalents of aqueous $NaIO_4$ were added. The mixture was stirred for 1 hour at room temperature and then quenched with aqueous $Na_2S_2O_3$. HPLC purification yielded **20** (~57% yield). 6 mg of **20** was dissolved in 1 mL of dichloromethane and treated with 1 mL of TFA. The solution was stirred at room temperature for 2 h to yield **12** (100% yield). For all purification steps, semi-preparative RP-HPLC equipped with a Phenomenex Luna $C_{18}$ column (20 to 100% acetonitrile in water with 0.01% TFA for 30 min, 4 mL/min, 1 min collection interval) was used.

### Stereochemical assignment of natural 12

Wildtype *K. oxytoca* ATCC 8724 was cultured aerobically in 3 mL of M9 minimal medium supplemented with 5 g/L casamino acids and 0.4% D-Gal at 30 °C for 24 h at 250 rpm and then dried at room temperature under reduced pressure (Genevac). The dried extract, 0.5 mg of the **12S** standard, and 0.5 mg of the **12R** standard were then individually resuspended in 200 μL of $H_2O$ and 400 μL of 1% Marfey's Reagent (1-fluoro-2-4-dinitrophenyl-5-L-alanine amide) along with 80 μL of 1 M $NaHCO_3$. The mixtures were left at 40 °C for 1 hour before addition of 40 μL of 2 M HCl. At this point, the mixtures were centrifuged for 10 min at 18,000 g, and 3 μL of the supernatants were subjected to HPLC-QTOF-MS analysis, as described above.

### RNA sample preparation and RNA-seq analysis

A Δ*pyr K. oxytoca* glycerol stock was streaked onto an LB-Agar plate supplemented with spectinomycin (50 μg/mL) and incubated aerobically at 30°C for 24 h in a stationary incubator. Single colonies were grown for 24 h aerobically in 5 mL of M9 minimal medium supplemented with 5 g/L casamino acids and 0.4% D-galactose at 30 °C and 250 rpm. Each Δ*pyr K. oxytoca* overnight culture was diluted 1:1000 into fresh medium (5 mL) and incubated aerobically at 30 °C and 250 rpm to mid-exponential growth ($OD_{600} = 0.8$). These conditions abrogated both detectable pyrazine and pyrazinone production. At this point, 10 μM of compound in DMSO, were added to the cultures; an equal volume of DMSO was added to cultures without compound as a solvent vehicle control. The cultures were incubated under aerobic conditions with the compounds for one hour at 30 °C and at 250 rpm. After incubation, the cultures were centrifuged, the supernatants were decanted, and the cell pellets were quickly resuspended in 5 mL of Bacterial RNAProtect (Qiagen). The samples were centrifuged again, and the pellets were frozen on dry ice and stored at −80 °C until RNA

extraction. The processed cell pellets were thawed on ice and resuspended in 200 μL of lysis buffer (10 mM Tris, 1 mM EDTA, 5 mg/mL lysozyme, 12.5 μg/mL proteinase K [pH 8.0]) and incubated at room temperature for 10 min. The RNA was extracted via the Qiagen RNeasy Mini kit, treated with TURBO DNase in a total volume of 100 μL at room temperature for 30 min, and extracted again using an RNeasy Mini kit. RNA-seq analysis was conducted by the Yale Center for Genome Analysis. Sequences were aligned to the *K. oxytoca* ATCC 8724 genome (NC_016612.1). Differential expression analysis was performed through DESeq2 and pathway enrichment analysis was performed through the Funage-Pro web-based platform.

### Human GPCR activity analyses

The PRESTO-Tango analyses were performed as previously published[81]. Briefly, HTLA cells, a HEK293 cell line that stably expresses β-arrestin-TEV and tTA-Luciferase (a kind gift from Gilad Barnea, Brown University), were plated in 96-well tissue culture plates (Eppendorf) in DMEM containing 10% FBS and 1% Penicillin/Streptomycin. One day after plating (after reaching approximately 90% confluence), 200 ng per well GPCR-Tango plasmids[37] in 20 μL DMEM were individually mixed with 400 ng polyethylenimine (Polysciences) in an equal volume of DMEM and incubated for 20 min at room temperature before adding the transfection mixture to the HTLA cells. 16-24 h after transfection, medium was replaced with 180 μL fresh DMEM containing 1% Penicillin/Streptomycin and 10 mM HEPES and 20 μL of **3** or **3-Homo** to reach a final concentration of 100 μM. Supernatants were aspirated 20 h after stimulation and 50 μl per well of Bright-Glo solution (Promega) diluted 20-fold with PBS containing 20 mM HEPES was added into each well. After 15 min incubation at room temperature, luminescence was quantified using a Spectramax i3x microplate reader (Molecular Devices). Activation fold for each sample was calculated by dividing relative luminescence units (RLU) for each condition by RLUs from PBS control. Dose response curves of each chemical were performed in a similar way with varying the ligand concentration from $10^{-4}$ M to $10^{-15}$ M.

### Phylogenetic analyses

The amino acid sequence of the *K. oxytoca* Pyr protein was used to BLAST "All genomes" on the DOE Integrated Microbial Genomes & Microbes (MG) database with an E-value cutoff of 1e-5. The resulting list of 459 homologs from this database encompassed 26 species of bacteria. Representative strains of the 26 species were selected and their operon architectures were plotted on a phylogenetic tree constructed with phyloT.

The sequence similarity network was generated via the Enzyme Similarity Tool[67] using the amino acid sequence of the *K. oxytoca* Pyr protein with an E-value cutoff of 1e-50 and an alignment score of 100. The network was visualized with Cytoscape and nodes with >90% sequence homology were clustered.

### Heterologous expression

The *pyr* gene from *K. oxytoca* ATCC 8724 was codon optimized for *E. coli* expression and ordered on a pet28a vector (C-terminal $His_{6x}$-tag fusion) from Twist Bioscience. The vector was transformed into *E. coli* BL21(DE3). Briefly, *E. coli* BL21(DE3) was cultured in LB medium overnight at 37 °C and at 250 rpm. This starter culture was sub-cultured 1:1000 into 3 mL of fresh LB, and the new culture was grown at 37 °C at 250 rpm until $OD_{600}$ of 0.6. 1 mL of this culture was then transferred to a 1.5 mL Eppendorf tube, and the tube was centrifuged for 1 min at 18,000 g. The supernatant was discarded, and the pellet was washed with 1 mL of ice-cold water after which the tube was centrifuged again. The pellet was washed a second time and then resuspended in 50 μL of ice-cold water. 1 μL of a 100 ng/μL solution of the pet28a vector containing the *pyr* gene was added to the resuspended cells and the solution was loaded into a 1 mm electrocuvette and pulsed at 1800 V,

25 μF, 200 Ω (BTX Gemini X2). Cells recovered overnight at 37 °C in LB broth and were plated on LB agar plates supplemented with kanamycin (50 μg/mL) at 37 °C.

For MS-based determination of the *pyr* products, *E. coli* BL21(DE3) containing the *pyr* pet28a vector was cultured, in triplicate, in 5 mL of LB medium with kanamycin (50 μg/mL) supplementation. For comparison, an LB medium control and LB cultures of *E. coli* BL21 (DE3) containing an empty pet28a vector were also generated and treated with kanamycin. The cultures were incubated under aerobic conditions at 37 °C and 250 rpm until they reached an $OD_{600}$ of 0.6–0.8. At this point, 100 μM of IPTG was added to the cultures and controls and the samples were incubated at 20 °C at 250 rpm for an additional 48 h. The samples were then dried at room temperature under reduced pressure (Genevac). The dried samples were resuspended in 200 μL of 1:1 methanol:water and 3 μL of the samples were subjected to HPLC-QTOF-MS analysis, as described above.

### Statistical analyses
For all statistical analysis and curve fitting, the software GraphPad Prism was used. For determining statistical significance, two-tailed unpaired t-tests were performed. * indicates $p < 0.05$; ** indicates $p < 0.01$; *** indicates $p < 0.001$.

### Reporting summary
Further information on research design is available in the Nature Portfolio Reporting Summary linked to this article.

## Data availability
All data supporting the findings of this study are available within the manuscript and its Supplementary Information. The source data generated in this study have been deposited in the Figshare database under accession code https://doi.org/10.6084/m9.figshare.25301908. The RNA-seq data generated in this study have been deposited in the NCBI database under accession code PRJNA1124412. The metabolomics data generated in this study have been deposited in the MassIVE database under accession code MSV000095925. Source data are provided in this paper.

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

## Acknowledgements

This work was supported by the National Institute of General Medical Sciences (1RM1GM141649-01 to J.M.C. and N.W.P.). R.H. was supported by a Ford Foundation Pre-Doctoral Fellowship and the National Institutes of Health Chemistry-Biology Interface Pre-Doctoral Training Grant program (T32GM067543). We thank Dr. Joonseok Oh for his suggestions and technical help with NMR spectroscopy and natural product isolation. RNA sequencing and transcriptomic analysis were performed as a fee-for-service at the Yale Center for Genome Analysis, which is supported by its high-performance computing grant (1S10OD030363-01A1).

## Author contributions

R.H. and J.M.C. conceived the project. R.H. and K.W. performed the experiments and analyzed the results. D.S. and N.W.P. performed and analyzed the GPCR experiments. R.H. and J.M.C. wrote the manuscript.

## Competing interests

The authors declare no competing interests.
