## [Transparent Peer Review file · Nature Communications]

Mucosal sugars delineate pyrazine vs pyrazinone autoinducer signaling in *Klebsiella oxytoca*

Corresponding Author: Professor Jason Crawford

Version 0:

Reviewer comments:

Reviewer #1

(Remarks to the Author)

The authors present interesting new metabolites that are produced by *Klebsiella oxytoca* in response to media conditions and specifically the neuraminic acid, Neu5Ac. The structure elucidation work is solid with insights gleaned from multiple techniques. The authors also identify the corresponding biosynthetic gene cluster and show these are wide-spread though the consequences of this prevalence are not discussed. Finally, they show that the pyrazinones activate the histamine receptor H4. Overall, this study presents important insights and it can make a nice contribution to Nature Communication once the following are addressed:

- Lines 93-94: I suggest the authors elaborate on the correlation between leupeptin BGC and nosocomial infections: how large of a panel was sequenced? What were the Control groups? Other relevant details?
- Figure 1g: please also show a co-injection of authentic and natural 3 to rule out that the compounds adventitiously exhibit identical retention times.
- The prevalence of the pyrazine BGCs is interesting but it is unclear what it means. A few sentences on this would be helpful. Are they controlling similar behavior in other bacteria? Do the authors believe they are induced by similar signals?
- Is the response to the homoarginine variant physiological? Or is it more of a curiosity as *K. oxytoca* does not produce this variant? This should be clarified.

Reviewer #2

(Remarks to the Author)

In the manuscript from Hamchand et al., the authors address the role of carbohydrate consumption on pyrazine and pyrazinone signaling production and their downstream regulation of iron metabolism (pathogenicity) in *Klebsiella oxytoca* and pro-inflammatory receptors in human cells. The authors found that sialic acid, mucus component, induced leupeptin biosynthesis. Using metabolomics, they found 2 features derived from leupeptin, which they show to correspond to two pyrazinones (3,4), which seem relevant in the production of iron metabolism, specifically the siderophore yersiniabactin (considered a virulence factor), and in the activation of the pro-inflammatory human histidine receptor H4 (HRH4). The authors also found that the general consumption of carbohydrates, namely galactose, induced a previously undescribed set of features (5-9) classified as pyrazines that are also involved in regulating iron acquisition, including by inducing the production of enterobactin. Finally, the authors investigate the conservation of pyrazine production across several species from different families.

This is an interesting topic of research with interesting finding for several fields of biology. The experiments included in the manuscript are well designed and clear. The chain of events described here are complex and can be difficult to follow in some cases. Having a unifying schematic of the paper would help to grasp all the events happening. Additionally, to improve some sections of the manuscript, and to help integrate of the new findings with current literature related to bacterial signaling, there are a few points to take into consideration.

Major Comments:

1. As the authors write, pyrazines 5 and 7 relate to previously reported autoinducer quorum sensing signals DPO and AI-3 in *Vibrio cholerae* and *E. coli*, which depend on threonine dehydrogenase encoded by *tdh*. Therefore, it is important to also determine here the impact of a *tdh* mutant in the production of the reported pyrazines. Namely, because *tdh* seems to have an important role in the production of this novel class of quorum sensing signals, therefore it is important to determine its role in the context of the molecules described in the current manuscript to help interpret the novel findings reported here in the context of previous literature on the role of *tdh* in quorum sensing signaling.
2. The authors observe that mucin induced the expression of leupeptin. They also showed that N-Acetylneuraminic acid – Neu5Ac, a predominant sialic acid found in mucin, could also activate this expression and that this induction was abolished in a *nanA* mutant. But can mucin still induce leupeptin expression in a *nanA* mutant? It would be important to understand if mucin induction was specific for sialic acid, or if other capping sugars, like fucose (also found in the mucus) could also do this induction.
3. Fig.1 shows that *nanA* mutant abrogates the production of feature 1, but it is not clear if feature 3 and 4 also require *nanA*. Moreover, it is written that it is shown in Fig.1g that feature 3 is produced in the presence of sialic acid, but there is no control showing that feature 3 is not produced when there is no sialic acid. In other words, it is important to show that the formation of 3 and 4 is present only Neu5Ac is metabolized by a *nanA* dependent pathway (and not in a *nanA* mutant). Overall, it is important to provide additional support for the claim that features 3 and 4 derive from an *leup* dependent pathway.
4. A final figure with a scheme of the pathways identified here and what they regulate would be very helpful in helping the reader understand the proposed findings reported here and this will also help the reader to better understand the follow of the different experiments.

Minor Comments:

1. In the text it is mentioned that Neu5Gc suppresses *K. oxytoca* growth. What do you mean by suppressing growth here? Sustained growth? Where is this data shown? Additionally, it is mentioned that Neu5Gc is absent in humans, but present in non-human models (which would make them non-suitable for testing this gene cluster), but here induction of leupeptin was tested and observed in (non-human) porcine mucin, with no apparent effect on *K. oxytoca* growth, which contradicts the previous statement. Please clarify.
2. Fig3 describes the role of PLP-enzyme for the production of pyrazines (features 5-9). From Fig.3a it is clear that features 5 and 7 are different from 6, 8, and 9. Are there functional differences between these 2 groups of compounds in terms of siderophore production? The authors show that complementation of the *pyr* mutant with feature 7 could induce siderophore production (Fig4). What about the features from the other group (6, 8, and 9)?
3. Features 5 and 7 seem to be dependent on *Tdh* (which also affects other signals like DPO or AI-3 production). As already mentioned above a *tdh* mutant should be analyzed. A *tdh* mutant should compromise the production of features 5 and 7, but not 6, 8, and 9, which could be an alternative way to show the role of 6, 8, and 9.
4. In Fig.4 is shown that a *leup* mutant is impaired in the production of feature 10 (yersiniabactin), as opposed to a *pyr* mutant. Isn't this contradictory with the data shown in Extended Fig. 3c where it is shown a reduction in 10a and 10b features in a *pyr* mutant?
5. In Fig.6 is described the conservation of the *pyr* genes across different families of bacteria. What about the leupeptin pathway?
6. In the introduction the authors refer that leupeptin pathway is associated with pulmonary infections, while tilivalline and tilimycin with colitis. In the RNA-seq data, with the complementation with feature 3 and 7, is there any indication whether this pyrazinones/pyrazines can also regulate tilivalline and tilimycin?
7. Line 178. Instead of "regulated" write enriched or increased.
8. In Fig.2C. It would help the reader if the pyrazine scaffold present in the molecules shown would be highlighted with a different color.
9. In Fig. 3a, to help the reader add the names of the compounds in green and in blue to the figure.

Reviewer #3

(Remarks to the Author)

(Remarks to the Author)

The primary purpose of your review is to provide feedback on the soundness of the research reported. This will help authors to improve their manuscript and editors to reach a decision. When composing your report, the following questions might assist you in writing a well-justified review, but please feel free to raise any further questions and concerns about the paper.

- What are the noteworthy results?
- Will the work be of significance to the field and related fields? How does it compare to the established literature? If the work is not original, please provide relevant references.
- Does the work support the conclusions and claims, or is additional evidence needed?
- Are there any flaws in the data analysis, interpretation and conclusions? Do these prohibit publication or require revision?
- Is the methodology sound? Does the work meet the expected standards in your field?
- Is there enough detail provided in the methods for the work to be reproduced?

This is a manuscript by Hamchand and coworkers entitled “Mucosal sugars delineate pyrazine vs pyrazinone autoinducer signaling in *K. oxytoca*” that uses a beautiful combination of chemical syntheses, microbial genomics, biochemistry and metabolomics to reveal how mucosal sugars drive complex biochemistry in the organism to advance its pathogenesis. This manuscript will advance the fields thinking in about host-pathogen interactions and provides a variety of new tools to help advance this important area. The paper is extremely well written and takes the reader through a complex landscape; the paper could have been five excellent manuscripts. However, this reviewer appreciates the completeness of the story and the many different biological outcomes of the host-pathogen interactions.

1. Noteworthy results and significance: specific mammalian mucosal sugars drive the production of pathogenic small molecule phenotypes – both previously identified metabolites and new metabolites. The biochemistry of these metabolites both in the formation and then biological activity is unique; the rigorous characterization is a breath of fresh air for our field and sets a new standard that should be followed. The use of new chemical synthesis of these metabolites will allow for analogues to be produced.

Food for thought: the authors mention that Neu5Gc suppressed *K. oxytoca* growth? Do the authors think that this is biologically important. Those individuals who eat red meat will be exposed to Neu5GC, as cattle have this pathway. Would this suggest that a specific diet could help certain individuals become resistant to *K. oxytoca* infection? Or those that harbor a mycobacterium infection – as those bacteria also have glycoyl containing sugars?

2. The work supports the conclusion and claims. The authors do not overextend their data. I like the biochemistry analysis of the GPCRs and understand that the authors could have extended to mammalian models but that would be TOO much for this already jammed packed manuscript.

3. There are no flaws in the data analysis. I would ask that the authors mine their HRMS data a bit further for all metabolites assigned key numbers in either a table or a figure. For these data, could they please show the MS envelope for each key M/Z – (i.e normal isotopic abundance and the ppm difference from predicted to shown). I believe that this should be easy for the authors and only enhance the rigor of their work.

4. Methodology: I was highly impressed with the rigor of each of the methods from chemical synthesis to bacterial genomics. The biochemistry is also sound. The authors should be commended for their work.

5. I have a few minor concerns/points of clarification for the methods and a few suggestions for figure edits to clarify items for the reader. This is well written manuscript and I only help to enhance with this constructive criticism:

a. The authors need a summary figure at the end of the manuscript to show how all the molecular programs work in concert – especially give the biochemical complexity. I think that this summary figure would illuminate the key findings more clearly for the reader/authors.

b. Pg 6 – “Variety of media” – could the authors comment on how many types and how these were chosen. Perhaps a table in the Leupeptin detection assay method section describing all the components of the 12 medias used – major reason why that media was used, would help others in rationalizing future media screens.

c. Preliminary mass spec – page 7; line 130 – is this HRMS? I believe it is and the authors should state as such in the main text. Please show the M/Z isotopic envelope for all compounds mentioned in this paragraph.

d. Figure 1f – show the mechanism of formation in the figure – those that do not have a chemical background might have a hard time following “proteolytic cleavage, cyclization and oxidation ...”

e. Page 11, line 170 – a reference for “molecular networking” would help the reader learn about this technique.

f. Page 11, line 187 – HPLC to purify and characterize – did the authors limit the detection to just UV? Or where other detection methods utilized?

g. Page 12, for the isolated compounds – could the authors state yield from the 5 L culture? Was the media supplemented with an optimized concentration of Neu5Ac? If so, how was the concentration of Neu5Ac chosen?

h. Very small – but please use small caps for D and L when describing stereochemistry.

i. Extended Figure 4c – state in the legend that started from either d or l Y.

j. Could the authors comment on the utility of RNAseq in bacteria compared to eukaryotes. Where the bacteria synched in growth phase? How long were the bacteria cultured (i.e. number of life cycles)? How could these parameters effect outcome?

k. Page 25 – what human cells express with histamine receptor HRH4?

l. Figure 6c – state in the figure caption that the background host is *E. coli* for the expression of the pyr.

m. Figure 1 – please label Leupeptin, etc in the figure – in addition to the number. This will help the reader to quickly find the molecules of interest.

Version 1:

Reviewer comments:

Reviewer #1

(Remarks to the Author)

The authors have provided a nice response to my prior comments (and those of other reviewers). I think the manuscript is now ready to be accepted for publication in Nat. Commun.

Reviewer #2

(Remarks to the Author)

We are happy with the way the authors have addressed our comments in the response letter and in this revised version of the manuscript. We think that the proposed additions have significantly improved the quality of the manuscript.

We have only some minor comments mainly related to the presentation and the scheme in Figure 7.

- We think it is important that the names of all the compounds shown in the figures appear in the figure legend, at least the first time that these are mentioned, when the names are not included in the figure. For example, the names of compounds 12 and 13, should be listed in the legend of Fig. 3.

The overall scheme in Fig. 7 is a great improvement; however, the figure and figure legend can still be improved.

Namely:

- The names of the compounds shown in the figure that are not in the figure should be listed in the legend, namely compounds 3, 12, and 13. Given the importance of compound 3, it would probably be important that its name is in the figure.

- Where it says "Intracelluer space", should say "Bacterial intracellular space" (or cytoplasm). Maybe you could have different background colors for the areas that are referring to the bacterial space, versus intestinal lumen.

- On the right-hand side of the figure, there is an arrow connecting Threonine to Pyrazines 5-9, but according to the results of the paper, Tdh be involved in yielding 5 and 7, but not 6, 8 and 9.

- And also why would this arrow (Pyrazines 5-9) be pointing to feature 7?

- This particular section (blue) of the figure needs to be fixed. It could also be added to the figure legend that these results from a broadly conserved PLP-dependent enzyme (Pyr, pyrazine biosynthesis protein) and Tdh (threonine dehydrogenase).

- In the figure legend it should say: "b. Overview of pathways for pyrazinone versus pyrazine signal production and signaling consequences in *K. oxytoca*. The light-orange section of the figure (left) denotes the consequences of Neu5Ac metabolism for the production of Pyrazinone signals" (is this correct? This part is only pyrazinone? The arrow is only pointing to compound 3, which is a pyrazinone), "...while the light-blue section (right) shows the consequences of general carbohydrate metabolism for production of pyrazine signals."

- The authors wrote in the figure (orange side) that Neu5Ac leads to both Pyrazine and Pyrazinone signaling but only show compound 3, which is a pyrazinone, is there a pyrazine missing here? Or the orange part should be referring to the impact of Neu5Ac on Pyrazinone signaling? If so change Mucosal Signaling to Pyrazinone Signaling.

Reviewer #3

(Remarks to the Author)

Reviewer 1:

The authors present interesting new metabolites that are produced by *Klebsiella oxytoca* in response to media conditions and specifically the neuraminic acid, Neu5Ac. The structure elucidation work is solid with insights gleaned from multiple techniques. The authors also identify the corresponding biosynthetic gene cluster and show these are wide-spread though the consequences of this prevalence are not discussed. Finally, they show that the pyrazinones activate the histamine receptor H4. Overall, this study presents important insights and it can make a nice contribution to Nature Communication once the following are addressed:

- Lines 93-94: I suggest the authors elaborate on the correlation between leupeptin BGC and nosocomial infections: how large of a panel was sequenced? What were the Control groups? Other relevant details?

Thank you for the suggestion. We elaborated on the previous panel which examined the prevalence of leup in clinical *K. oxytoca* isolates. The text now reads:

“We previously examined a panel of 84 clinical *K. oxytoca* isolates for the presence of either the *leup* operon or *npsB*, an NRPS of the tilimycin/tilivalline gene cluster.¹² We found that the presence of the leupeptin BGC in *K. oxytoca* correlates with nosocomial respiratory infections in this panel (14 out of 19 lung isolates, 2 out of 23 stool isolates), while the *npsB* gene correlates with intestinal tract dysfunction as expected (21 out of 23 stool isolates, 3 out of 19 lung isolates).¹² While the roles of tilivalline and tilimycin in hemorrhagic colitis are well studied,^{8,9} the operon for leupeptin biosynthesis is “silent” in *K. oxytoca* under typical laboratory culture conditions (i.e., leupeptin production is undetectable).”

- Figure 1g: please also show a co-injection of authentic and natural **3** to rule out that the compounds adventitiously exhibit identical retention times.

The green trace in the figure is a co-injection of synthetic **3** with the extract; however, given that another reviewer was also confused with our labeling, we changed the text in the figure to now say “*K. oxytoca* M9+Neu5Ac + **3** Co-injection.” Furthermore, we also added tandem MS spectra of synthetic and natural **3** to the SI to further validate that the compounds are the same.

- The prevalence of the pyrazine BGCs is interesting but it is unclear what it means. A few sentences on this would be helpful. Are they controlling similar behavior in other bacteria? Do the authors believe they are induced by similar signals?

While the presented pyrazines regulate iron acquisition systems in *K. oxytoca*, we are uncertain if they affect the same pathways in other bacteria or have broader, family dependent, bioactivities. To conservatively clarify our stance, lines 460-462 now state:

“Notably, while Pyr-derived pyrazines regulate iron acquisition systems in *K. oxytoca*, additional studies are required to determine their broader biological functions across bacterial lineages.”

- Is the response to the homoarginine variant physiological? Or is it more of a curiosity as *K. oxytoca* does not produce this variant? This should be clarified.

Because 3-homo is produced by related leup operons, we included this structure as preliminary SAR to 3. 3-homo is produced by an entomopathogen, so the response is not physiologically relevant to human GPCR signaling.

Reviewer 2:

In the manuscript from Hamchand et al., the authors address the role of carbohydrate consumption on pyrazine and pyrazinone signaling production and their downstream regulation of iron metabolism (pathogenicity) in *Klebsiella oxytoca* and pro-inflammatory receptors in human cells. The authors found that sialic acid, mucus component, induced leupeptin biosynthesis. Using metabolomics, they found 2 features derived from leupeptin, which they show to correspond to two pyrazinones (3,4), which seem relevant in the production of iron metabolism, specifically the siderophore yersiniabactin (considered a virulence factor), and in the activation of the pro-inflammatory human histamine receptor H4 (HRH4). The authors also found that the general consumption of carbohydrates, namely galactose, induced a previously undescribed set of features (5-9) classified as pyrazines that are also involved in regulating iron acquisition, including by inducing the production of enterobactin. Finally, the authors investigate the conservation of pyrazine production across several species from different families.

This is an interesting topic of research with interesting findings for several fields of biology. The experiments included in the manuscript are well designed and clear. The chain of events described here are complex and can be difficult to follow in some cases. Having a unifying schematic of the paper would help to grasp all the events happening. Additionally, to improve some sections of the manuscript, and to help integrate of the new findings with current literature related to bacterial signaling, there are a few points to take into consideration.

Thank you. We now include Figure 7 that presents a model and unifying schematic of the findings.

Major Comments:

1. As the authors write, pyrazines 5 and 7 relate to previously reported autoinducer quorum sensing signals DPO and AI-3 in *Vibrio cholerae* and *E. coli*, which depend on threonine dehydrogenase encoded by *tdh*. Therefore, it is important to also determine here the impact of a *tdh* mutant in the production of the reported pyrazines. Namely, because *tdh* seems to have an important role in the production of this novel class of quorum sensing signals, therefore it is important to determine its role in the context of the molecules described in the current manuscript to help interpret the novel findings reported here in the context of previous literature on the role of *tdh* in quorum sensing signaling.

Thank you for the comment; this is a fair point. To address this, we constructed a Δtdh *K. oxytoca* strain and employed our metabolomics pipeline to determine changes in pyrazine abundances between wildtype and Δtdh *K. oxytoca* strains. We found that the production of pyrazines **5** and **7**, which we propose are both products of threonine metabolism, is significantly reduced in the *tdh* mutant. Furthermore, the abundance of pyrazines **6**, **8**, and **9**, which we do not believe result from threonine metabolism, is not significantly different between wildtype and Δtdh *K. oxytoca* strains. This data has been added as **Extended Data Figure 3d** and the following paragraph was inserted after line 276:

“To determine the role of threonine metabolism on pyrazine production and biosynthesis, we constructed a *tdh* mutant strain ($\Delta tdh::spec$) and assessed relative pyrazine levels between the mutant and wildtype strains. We found that only production of pyrazines **5** and **7**, which incorporate *tdh*-derived aminoacetone, were significantly reduced in the Δtdh strain, while the abundances of pyrazines **6**, **8**, and **9** were not significantly different (**Extended Data Fig. 3d**). These findings support that threonine metabolism only partially contributes to the pyrazine network via aminoacetone substrate supply in *K. oxytoca*.”

2. The authors observe that mucin induced the expression of leupeptin. They also showed that N-Acetylneuraminic acid – Neu5Ac, a predominant sialic acid found in mucin, could also activate this expression and that this induction was abolished in a *nanA* mutant. But can mucin still induce leupeptin expression in a *nanA* mutant? It would be important to understand if mucin induction was specific for sialic acid, or if other capping sugars, like fucose (also found in the mucus) could also do this induction.

To address this concern, we performed a screen where we analyzed extracts of the *nanA* mutant cultivated in mucin medium, the *nanA* mutant cultivated in M9+Neu5Ac, the wildtype cultivated in mucin medium, and the wildtype cultivated M9+Neu5Ac, for the production of leupeptin. We found that only the wildtype was capable of producing leupeptin. The relative abundance of leupeptin was much greater in the M9+Neu5Ac experiments as anticipated than in the wildtype in mucin medium experiments. This finding supports our claim that Neu5Ac specifically induced leupeptin biosynthesis rather than other sugars in mucin. A plot showing this data has been added to the SI and a sentence has been added after line 144:

“Notably, the *nanA* mutant was unable to produce **1** when cultivated in mucin medium, supporting the claim that Neu5Ac is a specific elicitor of the leup operon (Supplemental Information).”

3. Fig.1 shows that *nanA* mutant abrogates the production of feature 1, but it is not clear if feature 3 and 4 also require *nanA*. Moreover, it is written that it is shown in Fig.1g that feature 3 is produced in the presence of sialic acid, but there is no control showing that feature 3 is not produced when there is no sialic acid. In other words, it is important to show that the formation of

3 and 4 is present only Neu5Ac is metabolized by a nanA dependent pathway (and not in a nanA mutant). Overall, it is important to provide additional support for the claim that features 3 and 4 derive from a leup dependent pathway.

We re-analyzed the data from Figure 1e to check if pyrazinone **3** was produced in both the wildtype and nanA mutant strains when supplemented with Neu5Ac. We found that pyrazinone **3** was only produced in the WT strain, supporting our claim that **3** results from the *leup* operon. This data has been added to the SI. It is worth noting that compound **3** deriving from a leupeptin dependent pathway in an entomopathogen was established in our previous paper (Li et al, *ACIE*, 2020) where we identified the leup BGC and characterized the compounds it is responsible for (leupeptin **1**, lys-leupeptin **2**, pyrazinone **3**, **3-Homo**, and various leupeptin/pyrazinone derivatives).

4. A final figure with a scheme of the pathways identified here and what they regulate would be very helpful in helping the reader understand the proposed findings reported here and this will also help the reader to better understand the follow of the different experiments.

Thank you – this was a good suggestion made by multiple reviewers. A summary figure (Figure 7) describing the pathways affected in this paper has been added. Lines 476-477 now read:

“In this work, we highlight the identification, biosynthesis, and chemical signaling responses of differentially-regulated pyrazine and pyrazinone autoinducer systems in *K. oxytoca* (Fig. 7).”

Minor Comments:

1. In the text it is mentioned that Neu5Gc suppresses *K. oxytoca* growth. What do you mean by suppressing growth here? Sustained growth? Where is this data shown? Additionally, it is mentioned that Neu5Gc is absent in humans, but present in non-human models (which would make them non-suitable for testing this gene cluster), but here induction of leupeptin was tested and observed in (non-human) porcine mucin, with no apparent effect on *K. oxytoca* growth, which contradicts the previous statement. Please clarify.

Humans cannot make Neu5Gc due to inactivation of cytidine monophospho-N-acetylneuraminic acid hydroxylase; however, Neu5Ac and *predominately* Neu5Gc are abundant in almost all other mammals. Thus, it would be expected that pig mucin would incorporate some minor level of Neu5Ac. In our initial screening of sugars, there were several incidents where Neu5Gc supplemented cultures did not grow while all of the others did; this was the basis for the claim that Neu5Gc suppressed *K. oxytoca* growth. We agree with the reviewer that this should have been assessed quantitatively. To examine this phenomenon quantitatively, we have since measured growth curves of *K. oxytoca* cultured in M9 medium with Neu5Ac, Neu5Gc, or no sugar supplementation (data shown below, at 24 hrs cultures were diluted 1:1 and OD₆₀₀ was determined

as 2 times the recorded value). Importantly, all cultures in this experiment grew. We found that Neu5Ac-supplemented cultures grew fastest while the Neu5Gc cultures grew about as fast as non-supplemented cultures. All cultures had reached stationary phase after 24 hours. While Neu5Gc cultures did grow slower than Neu5Ac cultures, the term suppression was incorrect. Therefore, we deleted text around Neu5Gc suppression and have changed line 123 to now read:

“Unlike Neu5Ac, Neu5Gc was unable to induce the production of **1** (Supplemental Information).”

Data supporting the claim that Neu5Gc does not induce leupeptin **1** production has been added to the SI. The following growth analysis was not added to the study.

2. Fig3 describes the role of PLP-enzyme for the production of pyrazines (features 5-9). From Fig.3a it is clear that features 5 and 7 are different from 6, 8, and 9. Are there functional differences between these 2 groups of compounds in terms of siderophore production? The authors show that complementation of the pyr mutant with feature 7 could induce siderophore production (Fig4). What about the features from the other group (6, 8, and 9)?

To address this comment, we performed RNA-seq analysis feeding compound **6** and looked at regulated genes/operons. Similar to compound **7**, compound **6** supplementation increased the transcription of genes involved with iron acquisition; a volcano plot showing this data, along with a table listing specific genes, is now included in the SI. As such, lines 345-348 now read:

“Similarly, examining the role of pyrazine **6** on *K. oxytoca* found that supplementation of this compound also differentially regulated the transcriptome (201 genes with $P_{adj} < 0.05$; $abs(\log_2FC) > 1$) and enhanced the transcription of genes related to iron acquisition and iron metabolism (Supplemental Information).”

And lines 360-362 read:

“Notably, this response was not unique to pyrazine 7 as pyrazine 6 also exhibited increased enterobactin transcription levels relative to a DMSO control (Supplemental Information).”

3. Features 5 and 7 seem to be dependent on Tdh (which also affects other signals like DPO or AI-3 production). As already mentioned above a *tdh* mutant should be analyzed. A *tdh* mutant should compromise the production of features 5 and 7, but not 6, 8, and 9, which could be an alternative way to show the role of 6, 8, and 9.

A detailed response to this comment is shown above where we discuss our metabolomic analysis of the *tdh* mutant and the complementation of feature 6. That said, since *tdh* regulates threonine metabolism generally, it may be difficult to parse direct pyrazine effects from non-direct *tdh* effects via genetic analysis. As such, we opted for RNA-seq analysis of compound 6 which we had already purified and could directly compare to our other results.

4. In Fig.4 is shown that a *leup* mutant is impaired in the production of feature 10 (yersiniabactin), as opposed to a *pyr* mutant. Isn't this contradictory with the data shown in Extended Fig. 3c where it is shown a reduction in 10a and 10b features in a *pyr* mutant?

We do believe that the *pyr* mutant may exhibit a slight reduction in yersiniabactin production; this is shown in both Extended Fig. 3c and Fig. 4f. However, in the triplicate experiments performed in Fig. 4f, this slight reduction was not statistically significant ($p=0.1698$).

5. In Fig.6 is described the conservation of the *pyr* genes across different families of bacteria. What about the leupeptin pathway?

The conservation of the *leup* operon was examined in our previous paper (Li et al, *ACIE*, 2020), which led us to study the *K. oxytoca* pathway in this manuscript. A note to this has been added to the text. Line 437 now reads:

“Notably, the *leup* operon is more narrowly distributed than the *pyr* gene.¹²”

6. In the introduction the authors refer that leupeptin pathway is associated with pulmonary infections, while tilivalline and tilimycin with colitis. In the RNA-seq data, with the complementation with feature 3 and 7, is there any indication whether this pyrazinones/pyrazines can also regulate tilivalline and tilimycin?

The strain we worked with does not harbor the tilivalline/tilimycin pathway, therefore our RNA-seq data cannot offer any insights into the regulation of these compounds by the pyrazines/pyrazinones.

7. Line 178. Instead of “regulated” write enriched or increased.

The word regulated has been replaced with enriched in this sentence.

8. In Fig.2C. It would help the reader if the pyrazine scaffold present in the molecules shown would be highlighted with a different color.

The pyrazine scaffolds in Fig.2C are now shown in green.

9. In Fig. 3a, to help the reader add the names of the compounds in green and in blue to the figure.

Names have now been added for compounds in green (2-Amino-3-oxobutanoic acid and Aminoacetone) and blue (Carbohydrate).

Reviewer 3:

Thank you.

Reviewer 4:

The primary purpose of your review is to provide feedback on the soundness of the research reported. This will help authors to improve their manuscript and editors to reach a decision. When composing your report, the following questions might assist you in writing a well-justified review, but please feel free to raise any further questions and concerns about the paper.

- What are the noteworthy results?
- Will the work be of significance to the field and related fields? How does it compare to the established literature? If the work is not original, please provide relevant references.
- Does the work support the conclusions and claims, or is additional evidence needed?
- Are there any flaws in the data analysis, interpretation and conclusions? Do these prohibit publication or require revision?
- Is the methodology sound? Does the work meet the expected standards in your field?
- Is there enough detail provided in the methods for the work to be reproduced?

This is a manuscript by Hamchand and coworkers entitled “Mucosal sugars delineate pyrazine vs pyrazinone autoinducer signaling in *K. oxytoca*” that uses a beautiful combination of chemical

syntheses, microbial genomics, biochemistry and metabolomics to reveal how mucosal sugars drive complex biochemistry in the organism to advance its pathogenies. This manuscript will advance the fields thinking in about host-pathogen interactions and provides a variety of new tools to help advance this important area. The paper is extremely well written and takes the reader through a complex landscape; the paper could have been five excellent manuscripts. However, this reviewer appreciates the completeness of the story and the many different biological outcomes of the host-pathogen interactions.

1. Noteworthy results and significance: specific mammalian mucosal sugars drive the production of pathogenic small molecule phenotypes – both previously identified metabolites and new metabolites. The biochemistry of these metabolites both in the formation and then biological activity is unique; the rigorous characterization is a breath of fresh air for our field and sets a new standard that should be followed. The use of new chemical synthesis of these metabolites will allow for analogues to be produced.

Food for thought: the authors mention that Neu5Gc suppressed *K. oxytoca* growth? Do the authors think that this is biologically important. Those individuals who eat red meat will be exposed to Neu5GC, as cattle have this pathway. Would this suggest that a specific diet could help certain individuals become resistant to *K. oxytoca* infection? Or those that harbor a mycobacterium infection – as those bacteria also have glycoyl containing sugars?

Based on a previous reviewer's query, we have removed text regarding Neu5Gc growth suppression. Line 123 now reads:

“Unlike Neu5Ac, Neu5Gc was unable to induce the production of **1** (Supplemental Information).” Data supporting the claim that Neu5Gc does not induce leupeptin **1** production has been added to the SI.

2. The work is supports the conclusion and claims. The authors do not overextend their data. I like the biochemistry analysis of the GPCRs and understand that the authors could have extended to mammalian models but that would be TOO much for this already jammed packed manuscript.

Thank you for your kind words.

3. There are no flaws in the data analysis. I would ask that the authors mine their HRMS data a bit further for all metabolites assigned key numbers in either a table or a figure. For these data, could they please show the MS envelope for each key M/Z – (i.e normal isotopic abundance and the ppm difference from predicted to shown). I believe that this should be easy for the authors and only enhance the rigor of their work.

m/z envelopes of all numbered metabolites are now shown in the SI. These envelopes also have an insert where the [M+H]⁺ observed and theoretical *m/z* ratios are shown along with their error in ppm.

4. Methodology: I was highly impressed with the rigor of each of the methods from chemical synthesis to bacterial genomics. The biochemistry is also sound. The authors should be commended for their work.

Thank you.

5. I have a few minor concerns/points of clarification for the methods and a few suggestions for figure edits to clarify items for the reader. This is well written manuscript and I only help to enhance with this constructive criticism:

a. The authors need a summary figure at the end of the manuscript to show how all the molecular programs work in concert – especially give the biochemical complexity. I think that this summary figure would illuminate the key findings more clearly for the reader/authors.

We agree with your comments. A summary figure (Figure 7) describing the pathways affected in this paper has been added. Lines 476-477 now read:

“In this work, we highlight the identification, biosynthesis, and chemical signaling responses of differentially-regulated pyrazine and pyrazinone autoinducer systems in *K. oxytoca* (Fig. 7).”

b. Pg 6 – “Variety of media” – could the authors comment on how many types and how these were chosen. Perhaps a table in the Leupeptin detection assay method section describing all the components of the 12 medias used – major reason why that media was used, would help others in rationalizing future media screens.

In practice, the OSMAC approach is not rational, it is a strategy that natural product chemists employ to access the chemical diversity of microbes. We picked the media that are typically used in our lab, and through our Leupeptin-reporter system, were able to quickly screen them. The OSMAC approach ultimately was not helpful. We previously used host factors to activate pathways in host-associated bacteria (Crawford et al. 2010. *Curr. Biol*). Ultimately, applying an analogous approach testing mucins led to a promising hit in the current study.

c. Preliminary mass spec – page 7; line 130 – is this HRMS? I believe it is and the authors should state as such in the main text. Please show the M/Z isotopic envelope for all compounds mentioned in this paragraph.

Yes, high-resolution electrospray ionization mass spectrometry was used to determine the mass of all compounds in this study. *m/z* isotopic envelopes are now shown for all numbered metabolites in the SI.

d. Figure 1f – show the mechanism of formation in the figure – those that do not have a chemical background might have a hard time following “proteolytic cleavage, cyclization and oxidation ...”

We added a proposed route of formation in the SI, Supporting Figure S30.

e. Page 11, line 170 – a reference for “molecular networking” would help the reader learn about this technique.

The molecular networking was done through the GNPS molecular networking platform. A citation to this has been added.

f. Page 11, line 187 – HPLC to purify and characterize – did the authors limit the detection to just UV? Or were other detection methods utilized?

We used a combination of chromophore- and mass-directed isolation techniques to purify the compounds (see lines 191-192).

g. Page 12, for the isolated compounds – could the authors state yield from the 5 L culture? Was the media supplemented with an optimized concentration of Neu5Ac? If so, how was the concentration of Neu5Ac chosen?

Galactose was used to purify the pyrazines since the pyrazines are dependent on carbohydrate metabolism broadly, not Neu5Ac specifically. This helped to eliminate pyrazinones from our pyrazine purification efforts. In our original screening for leupeptin production, we found that 0.4% sugar supplementation was sufficient for high pyrazine production, therefore we utilized this concentration in the large-scale purification. The yield of compounds 6, 7, and 9-Me are listed on Line 633 (1.3 mg, 3.0 mg, and 0.7 mg, respectively). All details for the isolation are stated in the methods sections under “Large-scale purification and structural elucidation of pyrazines.”

h. Very small – but please use small caps for D and L when describing stereochemistry.

We have gone through the manuscript and corrected this formatting error. Thank you.

i. Extended Figure 4c – state in the legend that started from either d or l Y.

An additional sentence has been added to the caption that reads:

“Synthesis was performed separately with each stereoisomer of *N*-Boc-(L- or D-)phenylalanine.”

j. Could the authors comment on the utility of RNAseq in bacteria compared to eukaryotes. Where the bacteria synched in growth phase? How long were the bacteria cultured (i.e. number of life cycles)? How could these parameters effect outcome?

Cultures were synched in growth phase at $OD_{600} = 0.8$ and 10 μ M of compound or an equal volume of DMSO was added to the samples. The samples were grown for an additional hour at 30°C 250 rpm and the pellets were RNA protected and frozen until RNA extraction. All details for the design of the RNA-seq experiment are stated in the methods section under “RNA sample preparation and RNA-seq analysis.” Addition details on how long the bacteria were cultures have been added to this section. As such, lines 692-697 now read:

“A Δ *pyr* *K. oxytoca* glycerol stock was streaked onto an LB-Agar plate supplemented with spectinomycin (50 μ g/mL) and incubated aerobically at 30°C for 24 hours in a stationary incubator. Single colonies were grown for 24 hours aerobically in 5 mL of M9 minimal medium supplemented with 5 g/L casamino acids and 0.4% D-galactose at 30 °C and 250 rpm. Each Δ *pyr* *K. oxytoca* overnight culture was diluted 1:1000 into fresh medium (5 mL) and incubated aerobically at 30 °C and 250 rpm to mid-exponential growth ($OD_{600} = 0.8$).”

We do not comment on the comparison between bacteria and eukaryotes in this study.

k. Page 25 – what human cells express with histamine receptor HRH4?

Thank you, this was a good point to address. HRH4 is expressed on immune system cells like monocytes, eosinophils, dendritic cells, T cells, and natural killer cells. To highlight this, lines 408-410 now read:

“HRH4, in particular, is expressed by several cells of the immune system, including monocytes, eosinophils, dendritic cells, T cells, and natural killer cells, and is known to regulate immune functions.⁶⁶”

l. Figure 6c – state in the figure caption that the background host is *E. coli* for the expression of the *pyr*.

Thank you. We added this detail to the caption.

m. Figure 1 –please label Leupiptin, etc in the figure – in addition to the number. This will help the reader to quickly find the molecules of interest.

Thank you. Leupeptin is now labelled in Figure 1.

Reviewer 1:

The authors have provided a nice response to my prior comments (and those of other reviewers). I think the manuscript is now ready to be accepted for publication in Nat. Commun.

We thank you for your guidance in preparing this manuscript for publication in *Nature Communications*.

Reviewer 2:

We are happy with the way the authors have addressed our comments in the response letter and in this revised version of the manuscript. We think that the proposed additions have significantly improved the quality of the manuscript.

We have only some minor comments mainly related to the presentation and the scheme in Figure 7.

- We think it is important that the names of all the compounds shown in the figures appear in the figure legend, at least the first time that these are mentioned, when the names are not included in the figure. For example, the names of compounds 12 and 13, should be listed in the legend of Fig. 3.

Thank you for the Figure clarity recommendations.

Names for all compounds in the figures now appear in the figure legend when first mentioned. Compounds 12 and 13 have been named PALA – Phenylalanine-derived 5-aminolevulinic acid analog (ALA) – and TALA – Tyrosine-derived ALA analog – respectively. Moreover, pyrazines 5-9 have been termed TALA-AA-Py, TALA-Py, PALA-Py, PALA-TALA-Pyr, and PALA-Py, respectively.

- Where it says “Intracelluer space”, should say “Bacterial intracellular space” (or cytoplasm). Maybe you could have different background colors for the areas that are referring to the bacterial space, versus intestinal lumen.

Thank you, Figure 7 now says “Cytoplasm” instead of Intracellular space.

- On the right-hand side of the figure, there is an arrow connecting Threonine to Pyrazines 5-9, but according to the results of the paper, Tdh be involved in yielding 5 and 7, but not 6, 8 and 9.

This is correct, Tdh is involved in the biosynthesis of pyrazines 5 and 7, but not 6, 8, 9. To clarify this, the arrow is now labeled “Pyrazines”, instead of “Pyrazines 5-9”.

- And also why would this arrow (Pyrazines 5-9) be pointing to feature 7?

Thank you for the question. Pyrazine **7** is the product of heterodimerization between Tdh-derived aminoacetone and compound **12**, which is why we believe the arrow is appropriate. Additionally, earlier in the paper, we compared the transcriptional effects of Pyrazine **7** and Pyrazinone **3** on the *K. oxytoca* transcriptome. Since Pyrazine **7** was explicitly tested relative to Pyrazinone **3**, this figure highlights Pyrazine **7** and Pyrazinone **3**.

- This particular section (blue) of the figure needs to be fixed. It could also be added to the figure legend that these results from a broadly conserved PLP-dependent enzyme (Pyr, pyrazine biosynthesis protein) and Tdh (threonine dehydrogenase).

Additional context has been added to the caption of Figure 7. This caption now reads:

“Proposed model for differential pyrazine versus pyrazinone signaling in *K. oxytoca* human lung isolates. **a.** Proposed mechanism of pyrazines versus pyrazinones induction based on carbohydrate utilization. In the presence of mucin-degrading bacteria (left, light-orange section), mucosal sugars – specifically Neu5Ac – and luminal sugars are released into the environment. The metabolism of Neu5Ac leads to *leup*-derived pyrazinone signaling and *pyr*-derived pyrazine signaling. Notably, pyrazine signaling is not restricted to Neu5Ac metabolism but relies on carbohydrate metabolism more broadly. In the absence of mucin-degrading bacteria (right, light-blue section), *K. oxytoca* metabolizes luminal sugars, resulting solely in *pyr*-derived pyrazine signaling. **b.** Overview of pathways differentially impacted by pyrazine versus pyrazinone signaling in *K. oxytoca*. When *K. oxytoca* metabolizes Neu5Ac, the *leup* operon is activated leading to the production of leupeptin **1** and its pyrazinone derivative **3** (left, light-orange section). Leupeptin pyrazinone **3** can activate host HRH4, potentially causing an inflammatory response, and enhances production of the siderophores yersiniabactin **10** and enterobactin, which scavenge for environmental iron. Yersiniabactin, specifically, can act as a host virulence factor. Under broad carbohydrate metabolism conditions, including Neu5Ac, Pyr (pyrazine biosynthesis enzyme) couples L-Phe or L-Tyr with succinyl-CoA to form PALA **12** and TALA **13** (right, light-blue section). These compounds can dimerize with themselves, each other, or Tdh (threonine dehydrogenase)-derived AA to form pyrazines **5-9**. While all of these compounds may affect the *K. oxytoca* transcriptome, RNA-sequencing shows that PALA-AA-Py **7** stimulates enterobactin biosynthesis, promoting iron acquisition. The *leup* BGC, *ybt* BGC, and *ent* BGC refer to the leupeptin, yersiniabactin, and enterobactin biosynthetic gene clusters, respectively. Figure was created with BioRender.com.⁸¹”

- In the figure legend it should say: “b. Overview of pathways for pyrazinone versus pyrazine signal production and signaling consequences in *K. oxytoca*. The light-orange section of the figure (left) denotes the consequences of Neu5Ac metabolism for the production of Pyrazinone signals” (is this correct? This part is only pyrazinone? The arrow is only pointing to compound 3, which is a pyrazinone), “...while the light-blue section (right) shows the consequences of general carbohydrate metabolism for production of pyrazine signals.”

Thank you for the question. The light-orange section denotes the signaling consequences of leupeptin derived pyrazinone **3** while the light-blue section denotes the signaling consequences of

pyrazine 7. Neu5Ac metabolism leads to both pyrazinone and pyrazine signaling, but broad sugar metabolism leads only to pyrazine signaling. The caption of Figure 7 has been updated to make this clear:

“In the presence of mucin-degrading bacteria (left, light-orange section), mucosal sugars – specifically Neu5Ac – and luminal sugars are released into the environment. The metabolism of Neu5Ac leads to *leup*-derived pyrazinone signaling and *pyr*-derived pyrazine signaling. Notably, pyrazine signaling is not restricted to Neu5Ac metabolism but relies on carbohydrate metabolism more broadly. In the absence of mucin-degrading bacteria (right, light-blue section), *K. oxytoca* metabolizes luminal sugars, resulting solely in *pyr*-derived pyrazine signaling.”

- The authors wrote in the figure (orange side) that Neu5Ac leads to both Pyrazine and Pyrazinone signaling but only show compound 3, which is a pyrazinone, is there a pyrazine missing here? Or the orange part should be referring to the impact of Neu5Ac on Pyrazinone signaling? If so change Mucosal Signaling to Pyrazinone Signaling.

Thank you for the comment. The light-orange section reflects the consequences of Pyrazinone **3** signaling, though you are correct that Neu5Ac metabolism leads to both pyrazine and pyrazinone signaling. The caption to Figure 7 has been updated for clarity:

Reviewer #3 (Remarks to the Author):

Thank you for help in preparing this manuscript for publication to *Nature Communications*.